# Impact of Icebergs on Net Primary Productivity in the Southern Ocean

Shuang-Ye Wu[1,2], Shugui Hou[2]

[1]Department of Geology, University of Dayton, Dayton, Ohio 45469, USA
[2]School of Geographic and Oceanic Sciences, Nanjing University, Nanjing 210093

*Correspondence to*: Shuang-Ye Wu (swu001@udayton.edu), Shugui Hou (shugui@nju.edu.cn)

**Abstract.** Productivity in the Southern Ocean (SO) is iron-limited, and supply of iron dissolved from aeolian dust is believed to be the main source from outside the marine environment. However, recent studies show that icebergs could provide comparable amount of bioavailable iron to the SO as aeolian dust. In addition, small scale areal studies suggest increased concentrations of chlorophyll, krill, and seabirds surrounding icebergs. Based on previous research, this study aims to examine whether iceberg occurrence has a significant impact on marine productivity at the scale of the SO, using remote sensing data of iceberg occurrences and ocean net primary productivity (NPP) covering the period 2002-2014. The impacts of both large and small icebergs are examined in four major ecological zones of the SO: the continental shelf zone (CSZ), the seasonal ice zone (SIZ), the permanent open ocean zone (POOZ) and the polar front zone (PFZ). We found that the presence of icebergs is associated with elevated levels of NPP, but the difference vary in different zones. Grid cells with small icebergs on average have higher NPP than other cells in most iron deficient zones: 21% higher for the SIZ, 16% for the POOZ, and 12% for the PFZ. The difference is relatively small in the CSZ where iron is supplied from melt water and sediment input from the continent. In addition, NPP of grid cells adjacent to large icebergs on average is 10% higher than that of control cells in the vicinity. The difference is larger at higher latitudes, where most large icebergs are concentrated. From 1992-2014, there is a significant increasing trend for both small and large icebergs. The increase was most rapid in the early 2000s, and has levelled off since then. As the climate continues to warm, the Antarctic Ice Sheet is expected to experience increased mass loss as a whole, which could lead to more icebergs in the region. Based on our study, this could result in higher level of NPP in the SO as a whole, providing a possible negative feedback for global warming in near future.

## 1 Introduction

Iron, the fourth most abundant crustal element, has received widespread attention since the late 1980s due to its particular relevance for biogeochemical cycles in the ocean, especially in the high-nutrient low chlorophyll (HNLC) oceanic areas where plenty of macro-nutrients, such as nitrate and phosphate, are present, but with limited Fe (Falkowski et al., 1998). Changes in iron supply to oceanic planktons are thought to have a significant effect on concentrations of atmospheric carbon dioxide by altering rates of carbon sequestration, a theory known as the 'iron hypothesis' (Martin et al., 1990). The iron hypothesis stimulated new interests into the iron-enrichment experiments ranging from microcosmic (e.g. de Baar et al., 1990) to large

scale studies (e.g. de Baar et al., 2005) in either natural or artificial settings (Blain et al., 2007; Smetacek et al., 2012). These studies have demonstrated that Fe serves as a major control over both the primary productivity and the planktonic community structure in HNLC waters in more than 25% of the world oceans. Fe can take different forms in the oceans, e.g. particulate, colloidal, soluble, and as inorganic or organic complexes. The concentration of total dissolved Fe is generally low in the SO

water. de Baar and de Jong (2001) estimate it between 0.1–0.6 nM. Klunder et al. (2011) reported that within the upper surface mixed layer of the Atlantic sector of the SO, the concentrations of dissolved Fe vary between 0.1 and 0.3 nM. Klunder et al. (2014) also observed very low dissolved Fe concentrations (0.01–0.1 nM range) in the surface waters of the Weddell Sea, and within the Drake Passage polar regime. This low level of Fe is often insufficient to sustain the planktonic growth hence limits the primary productivity of the ocean (Price and Morel, 1998). The wide SO area is a place for intermediate- and deep-water

formation, as well as the largest repository of unused macronutrients in surface waters. As a result, it is a potential site for enhanced sequestration of carbon (Cooper et al., 1996) if extra Fe could be supplied to promote plankton growth. It is suggested that this is the primary reason for the SO $CO_2$ 'leak' to be stemmed during ice ages, increasing ocean $CO_2$ storage and thus lower atmospheric $CO_2$ concentration (Sigman et al., 2000).

Cassar et al. (2007) summarized five sources of bioavailable iron to the SO surface waters, i.e., melting of sea ice, the release

of dissolved iron or resuspension of sediments, upwelling supplies iron, vertical mixing supplies iron, and delivery of soluble iron by aerosol deposition supplies. They further confirmed that aerosol iron deposition has a significant influence on variability of net community production (NCP), as well as gross primary production (GPP) over the large SO areas. The first global maps of aerosol iron flux to the ocean were based on aerosol sampling networks, such as the SEAREX (Duce and Tindale, 1991). These maps revealed that the SO waters are characterized by particularly low aerosol iron fluxes. More

recently, German et al. (2015) demonstrate that seafloor venting may provide a significant source of bio-essential Fe to the oceans as the result of a close coupling between Fe and organic carbon in hydrothermal plumes. Particles in ice can be transported by icebergs from coastal regions into the SO, releasing bioavailable Fe to the open water during melting along the tracks. Thus icebergs in the SO waters have also been shown to transport nutrients, together with bioavailable Fe, which could have a significant impact on ocean primary productivity (Raiswell et al., 2008; Lancelot et al., 2009; Schwarz and Schodlok,

2009). Raiswell et al. (2008) demonstrated the presence of potentially bioavailable Fe as ferrihydrite and goethite in nanoparticulate clusters, and estimated comparable fluxes of bioavailable iron supplied to the SO from icebergs (0.06–0.12 Tg yr$^{-1}$) and aerosol deposition (0.01–0.13 Tg yr$^{-1}$). Raiswell and Canfield (2012) recently even suggested that icebergs could supply more than 90% of total colloidal and filterable Fe in the SO. Duprat et al. (2016) studied the influence of giant icebergs, over 18 km in length, on marine primary production in the SO, and detected substantially enhanced chlorophyll levels, typically

over a radius of at least 4–10 times the iceberg's length, that can persist for more than a month following passage of a giant iceberg. Therefore, an increase in the SO carbon sequestration through enhanced marine productivity due to fertilization by giant icebergs may become more important with future warming.

Despite the great interests and several small scale areal studies suggesting increased concentrations of chlorophyll, krill, and seabirds surrounding icebergs, there seems to be a lack of large-scale studies that examine the impact of icebergs on total

productivity in the SO. However, this is an important question, because in order for icebergs to have a significant role in the carbon cycle, hence serving as a possible negative feedback in the climate system, they should have an observable impact on NPP beyond the scale of individual icebergs. Therefore, this study aims to establish whether icebergs have a significant impact on marine productivity at the scale of the entire SO based on remote sensing data. In addition, most previous studies focus on

large icebergs (> 18.5 km long) (Duprat et al., 2016; Helly et al., 2011). Although these large icebergs may have a big impact on productivity of surrounding area, they are relatively rare compared to much more abundant small icebergs. Tournadre et al. (2008) estimated over 8000 small icebergs (< 1km) alone in one year. Moreover, small icebergs have a much larger surface to volume ratio, providing relatively more substrate for organisms (Smith et al., 2011). So when considered in total, they can have a big impact over much larger area in the SO. The majority of small icebergs come from the dislocation and breaking of

large icebergs, and therefore serve as an important diffuse process for transport of freshwater and nutrients including iron (Tounadre et al., 2016). In this study, we examine both large and small icebergs to determine if similar impacts on the ocean net primary productivity (NPP) could be observed at the SO scale. In addition, this study also examines the secular trend of icebergs to establish whether increased icebergs could be observed with present climate change. These are important questions based on which we can evaluate whether icebergs could provide an important negative feedback for global warming.

**2 Data and methodology**

**2.1  Data**

Three major datasets are used in this study. Monthly ocean NPP (mg C m$^{-2}$ day$^{-1}$) data are obtained from Oregon State University Ocean Productivity website (http://www.science.oregonstate.edu/ocean.productivity/index.php) and covers the years 2002-2014. The data are derived from MODIS R2013.1 data using the Vertically Generalized Production Model (VGPM)

(Behrenfeld and Falkowski, 1997) as the standard algorithm. In this model, NPP is a function of chlorophyll, available light, and the photosynthetic efficiency. This is a global dataset with a spatial resolution of 0.167 degrees.

Large icebergs are routinely tracked and monitored. The Brigham Young University (BYU) Centre for Remote Sensing produces and maintains an Antarctica Iceberg Tracking Database (http://www.scp.byu.edu/data/iceberg/database1.html) for icebergs with length larger than 5 km (Stuart and Long, 2011) since 1992, using six different satellite scatterometer

instruments. Icebergs are identified using enhanced resolution scatterometer backscatter images. The dataset contains the daily location for all identified icebergs. We summarize the track data into monthly 1x1 degree gridded format to facility the analysis with NPP data.

A monthly dataset of small iceberg (<3km) is obtained from Iceberg Database of the Merged Altimeter for Altiberg project for 1992-2014 (ftp://ftp.ifremer.fr/ifremer/cersat/projects/altiberg/). The data are generated based on the analysis of high

resolution altimeter waveforms from images of 9 satellite-based altimeters (Tournadre et al., 2016). The data contain three

variables: iceberg presence probability, surface area and volume. Iceberg probability of presence is defined in Tournadre et al. (2012) as "the ratio of the number of icebergs detected within a grid cell by the total number of valid satellite data samples within the same grid cell". It is essentially normalized iceberg frequency, and we shall refer to this variable as such to make it more explanatory. Details of how iceberg surface area and volume are estimated can be found in Tournadre et al. (2012). They

are in general correlated with iceberg frequency. The dataset covers the SO south of 40º S at a spatial resolution of 1 degree. For examining NPP-iceberg relationship, we extracted all datasets for the common time period 2002-2014 for our study area (defined below), and resampled them to a common grid of 1x1 degree. To establish iceberg trend, we use the full range of the datasets from 1992 to 2014.

## 2.2 Study area and delineation of ecological zones in the SO

Multiple sources of iron exist in the SO. In order to separate the impact of icebergs from some known sources, we divide our study area into four distinct ecological zones, each with different sources of nutrient supply (Figure 1). These zones were adapted and modified from previous studies (Treguer and Jacques, 1992; Moore and Abbott, 2000; Ito et al., 2005), and each zone with similar oceanographic and nutrient conditions. They are listed as follows:

- Continental shelf zone (CSZ): area less than 500 m in depth around the Antarctica continent and permanent ice shelves.
- Seasonal Ice Zone (SIZ): area north of the CSZ and within the mean maximum ice extent, defined by area with mean sea ice concentration greater than 70% in February. This region is delineated using the mean monthly sea ice 1979-2013 obtained from the National Snow and Ice (NIC) Data Centre (ftp://sidads.colorado.edu/pub/DATASETS/nsidc0192_ seaice_trends_climo/monthly-climatology/).

- Permanent Open Ocean Zone (POOZ): area north of the SIZ and south of the Polar front zone (defined below). This zone
is largely ice free throughout the year.

- Polar Front Zone (PFZ): area within 1 degree (~110km) of the polar front defined by Dong et al. (2006). This is an area characterized by strong upwelling and eddy mixing.

The Subtropical Front, northern boundary of the Antarctica Circumpolar Circulation, usually defines the border of the SO. However, since few icebergs drift past the PFZ, area of the SO north of the PFZ is not considered in this study.

**2.3 Methods to study the impact of small icebergs on NPP**

**2.3.1 Comparison of NPP in waters with and without icebergs**

We first calculate total annual primary production (Gt) by summing the area-weighted monthly production in each zone. Mean productivity (i.e. production per unit area) is calculated in two ways: total production over total area of the zone and over only productive waters (i.e. water that is not completely ice-covered and is with some light to allow biological production).

Productive water is identified as grids with NPP greater than zero, and its area varies month by month. Annual mean productivity is calculated as the sum of monthly mean productivity.

Based on the small iceberg data, we identify 1x1 degree grids with iceberg presence and those without on monthly timescale. For each zone, we calculate monthly total production, mean productivity over all area and mean productivity over productive waters for grids with and without icebergs respectively, based on which annual statistics are then compiled. Total iceberg volume is also calculated for each zone at both monthly and annual timescale. We compare both monthly and annual productivity for waters with and without icebergs.

### 2.3.2 Correlation between NPP and iceberg frequency

In order to further examine the quantitative relationship between NPP and small icebergs, we conduct the correlation analysis between the two factors. Normalized iceberg frequency is chosen over iceberg volume because it seems to better correlate with NPP. The possible reason is that for larger icebergs basal melting is small compared to their breaking into smaller icebergs (Tournadre et al., 2015). These smaller icebergs act as an important diffuse process for nutrient transport. Therefore, iceberg frequency may have a more direct impact on NPP than total iceberg volume. We calculate Pearson's correlation coefficient ($r$) for annual total net production and the annual mean iceberg frequency. The use of annual data eliminates the seasonal cycles that exist in both variables, which can artificially inflate the significance of correlation. Given that both variables are positively skewed, we also tried non-parametric rank correlation Spearman's *rho*. Both methods yielded similar results, with Spearman's *rho* giving slightly higher correlation than Pearson's *r*, indicating the existence of non-linear correlation. We perform the correlation analysis in two ways. First, we calculate the correlation coefficient between annual production and iceberg frequency at each grid point (local correlation), so that we can control it for spatially varied factors such as solar radiation, length of day, and ocean circulations. We then summarize the correlation coefficients by ecological zones. Second, we calculate a single correlation coefficient for annual productivity and iceberg frequency of all grid points within a zone (zonal correlation), and compare the strength of correlation between zones.

When analysing the impact of small icebergs on NPP, we did not include the large iceberg data for several reasons. First, the BYU large iceberg database is a tracking dataset, which only contains location information without any characteristics of the iceberg (such as length and volume). Therefore, it could not be incorporated into the correlation analysis. However, this is not a major problem, since large icebergs are so rare, occupying on average 157 grids per year compared to close to 7000 grids occupied by small icebergs within our study area. Their impact should not have a statistical significance on the analysis. Second, small icebergs are associated with large icebergs, and they are often found within a relatively short distance from large icebergs (Tournadre et al., 2015). Therefore, when we identify grids with iceberg presence, the exclusion of large icebergs does not make much difference in the results. Owing to these reasons, we excluded large icebergs when analysing the impact of small icebergs on the SO NPP.

### 2.4 Method to study impact of large icebergs on NPP

Since large icebergs are relatively rare, at certain point in time, they cover very small portion of the ocean surface. It is therefore not appropriate to adopt the same approach used in studying small icebergs. We take a different approach instead. We first

summarize the track data into monthly snapshots of 1x1 degree grid of iceberg counts. The great majority of the grids only get 1 large iceberg. Very rarely are more than one large iceberg present in a grid at a given month. For each iceberg occupied grid, we select a 7x7 degree (49 grids) window around it. Within the window, we identify all grids with icebergs (iceberg grids), grids immediately adjacent to iceberg grids (adjacent grids), and the rest of the grid within the window (nearby grids). We

calculate the mean NPP for each group for a single iceberg grid, and repeat the same operation for all iceberg grids at monthly timescale. We then compare the mean NPP of these three groups to see if they are significantly different. Pair-wise $t$ test is used to establish the statistical significance of the difference between groups. The difference is summarized by ecological zones, and their seasonal variations examined.

## 2.5 Temporal trend of icebergs and NPP in the SO

It is hypothesized that the positive impact of icebergs on the NPP could potentially serve as a negative feedback for climate change because warming could cause loss of the polar ice sheets through increased calving and other mechanisms leading to more icebergs. We use the complete small and large iceberg data from 1992 to 2014 to examine if any significant trend exists. We calculate the linear trend for the annual iceberg frequency for both large and small icebergs and total volume for small icebergs, and use the standard $F$ statistic to establish its statistical significance (Lomax, 2007). Since the NPP data are only

available for 2002-2014, we also calculate the linear trend for icebergs during this period and the corresponding trend in NPP. We calculate the iceberg and NPP trend for the entire SO as well as for each of the ecological zones. Moreover, previous studies suggest large regional variation in ice mass change for the Antarctica (Shepherd et al., 2012; Hanna et al., 2013; Sasgen et al., 2013). We therefore divide the SO into three sections: the Southern Atlantic (SA, from 70W to 30E), the Southern Indian (SI, from 30E to 150E), and the Southern Pacific (SP, from 150E to 70W) Ocean. We calculate the linear trend for icebergs in

each of the sections.

## 3 Results and discussion

### 3.1 Spatial distribution of NPP

Figure 2a shows the spatial distribution of mean NPP in the SO in relation to the defined ecological zones. Table 1 summarizes the area and total primary production in different ecological zones. Total production of each zone depends on productivity of

the water, total area of the zone, and the length of the time when the area is productive (i.e. the area is not ice covered and has some light to allow biological activities). If only productive water is considered, productivity is the highest in the CSZ ($113.5 + 3.7$ gC m$^2$ yr$^{-1}$) near the coast of Antarctica. It gradually decreases in the SIZ ($63.0 + 1.6$ gC m$^2$ yr$^{-1}$), reaching the minimum values in the POOZ ($57.5 + 1.0$ gC m$^2$ yr$^{-1}$). It then increases in the PFZ ($76.5 + 0.8$ gC m$^2$ yr$^{-1}$). Despite high productivity, zones at high latitudes have relatively low percentage of productive waters (17% for CSZ and 39% for SIZ)

because they are largely covered by ice and light-limited during the austral winter. Figure 3 shows the seasonal cycles of monthly mean productivity for all zones. The seasonal contrast is the greatest for high latitude zones such as the CSZ and the

SIZ when productivity could drop to zero during winter months. The POOZ and the PFZ on the other hand remain productive throughout the year, even though their per unit area productivity is not as high. Combining all the factors, on average the POOZ has the highest annual total production (0.63+0.010 Gt C), followed by the SIZ (0.40+0.010 Gt C) and the PFZ (0.38+0.004 Gt). The CSZ contributes 0.06+0.003 Gt C each year. Our results on the distribution of primary production and productivity are qualitatively comparable with previous studies (e.g. Treguer and Jacques, 1992; Moore and Abbott, 2000), although exact quantitative comparison is difficult given different methods and data based on which the ecological zones are delineated.

## 3.2 Spatial distribution of small icebergs

Figure 2b shows the spatial distribution of normalized frequency for small icebergs in the SO in relation to the defined ecological zones. Table 2 summarizes the area of all $1°x1°$ cells with iceberg presence and that of productive cells with iceberg presence in each zone, as well as annual total iceberg volumes in each zone. Iceberg frequency has three well-defined high value regions just off the coast of Antarctica, one in each ocean. This is largely caused by general iceberg circulation pattern in the SO (Tournadre et al., 2012; Gladstone et al., 2001). Most of these high frequency areas are confined in the SIZ, except for the largest one, located at southern Atlantic Ocean north of the Weddell Sea, which extends to the POOZ and the PFZ. Therefore, the SIZ has by far the largest area of grids with icebergs as well as total volume of icebergs. Even though the grid cells with iceberg presence only accounts for small portion of the total area in each zone, they constitute a much larger portion of the productive waters. For example, grids with icebergs account for 6% of total area but 32% of productive waters in the CSZ. They account for 16% of the total area but 36% of the productive waters in the SIZ. Since POOZ and PFZ are largely ice free all year round, the percentage is similar for total area and productive waters. Grids with icebergs only account for a small fraction of the area in both zones (9% for POOZ and 4% for PFZ) with relatively small quantity of icebergs (Table 2).

## 3.3 Impacts of small icebergs on NPP

We calculate the area-weighted average monthly NPP based on which annual NPP is compiled within each zone for grids with and without iceberg presence. Results are presented in Table 2. With the exception of the CSZ, where there is little difference between the two, NPP in grid cells with iceberg presence is significantly higher than that in cells without icebergs. The difference is most significant for the SIZ with iceberg cell NPP 21% higher than those without. It is 16% higher for the POOZ and 12% higher for the PFZ. With the exception of the CSZ, the differences in all other zones are statistically significant based on the two-sample Student's $t$ test. We also examine the seasonal variation of the difference in NPP between cells with and without icebergs (Figure 4). Figure 4 also shows the monthly total area of grid cells with icebergs. Iceberg presence also shows a distinct seasonal cycle with low numbers in the austral winter and high numbers in the austral summer. In the CSZ, there is virtually no difference in NPP between cells with and without icebergs for all months except December. In all other zones, NPP for cells with icebergs is consistently higher than that in cells without for most productive months in the austral spring, summer and fall. It is sometime lower, but mostly in the winter months when both productivity and iceberg presence are low.

Our results suggest that the presence of icebergs is associated with elevated level of productivity in most part of the SO, although their impact varies in different ecological zones. Possible explanations lie in the different nutrient sources and settings in these zones. Although short in duration, the productive season of the CSZ has the highest values of primary productivity. In this zone, nutrients are largely supplied by shelf sediments derived from the continent. Elevated iron concentration has also

been recorded near the coast (Johnson et al., 1997; Moore and Braucher, 2008). High levels of nutrients often lead to plankton bloom observed near the coast as well as downstream of islands (Blain et al., 2007; Pollard et al., 2009). In addition, Treguer and Jacques (1992) noted high levels of macronutrients at the beginning spring, but during summer these macronutrients become depleted in the coastal waters. This seems to suggest that primary production is limited by macronutrients instead of micronutrients such as iron. As a result, the presence of icebergs does not seem to have any significant impact on productivity

near the coast.

In both the CSZ and the SIZ, sea ice can supply additional iron. Lannuzel et al. (2010) suggest that iron is incorporated from the ocean into the ice during sea-ice formation, and hence the iron concentration within the sea ice can be an order of magnitude higher than in the underlying water (Lannuzel et al., 2007). However, some model studies show that the amount of iron released from sea-ice is minor compared to the sediment source (Lancelot et al. 2009; Wadley et al., 2014). This can partly explain the

generally much lower productivity in the SIZ, despite similarly high levels of macronutrients in the water during the productive season (Treguer and Jacques, 1992). In the SIZ, icebergs seem to have the largest impacts. The productivity in the grid cells with iceberg presence in general is 21% higher than those without icebergs. This large impact can also be attributed to the fact that the majority of icebergs occur in this zone: 65% of all iceberg grids and 73% of total iceberg volume occur in this zone, occupying 36% of the productive waters of this zone.

In contrast to the CSZ and the SIZ, where seasonal ice retreat often leads to stratified system (Sullivan et al., 1988), the POOZ is typically well-mixed and ice free. Mixing brings nutrient-rich water towards the surface, resulting in high supply of macronutrients in the surface water (Pollard et al., 2002). However, although nutrient rich, the POOZ is characterized by low levels of productivity. The annual mean productivity of the POOZ is the lowest among all four zones. It has been recognized that such production in this region is likely to be limited by the micronutrient iron, which plays an important role in chlorophyll

synthesis and hence phytoplankton growth (Geider and la Roche, 1994; Martin and Fitzwater, 1988; Martin et al., 1990). In this area, iron released from icebergs can be particularly important. About 30% of iceberg grids (19% by volume) occur in the POOZ, occupying about 9% of its productive waters. On average, they increased the productivity of these grids by 16%.

The Antarctic Polar Front is one of several strong fronts within the Antarctic Circumpolar Current. It is characterised by a strong gradient in the sea surface temperature within the PFZ, and marks the surface transition between cold Antarctic surface

water to the south and warmer sub-Antarctic surface waters to the north (Orsi et al., 1995; Moore et al., 1999; Dong et al., 2006). Meander-induced upwelling or increased eddy mixing may lead to increased fluxes of nutrients, including micronutrients such as iron, in the surface layer, particularly where the ACC interacts with large topographic features (Moore et al., 1999; Moore and Abbott, 2002). Several studies have suggested that higher levels of dissolved iron in surface waters at the PF led to elevated phytoplankton production (de Baar et al., 1995; Measures and Vink, 2001). Although only about 6% of

the iceberg grids (3% by volume) are found in the PFZ, occupying 4% of its productive grids, iceberg presence still have an observable impact, increasing the average productivity by 12%. This increase is lower than that observed in the SIZ and the POOZ, partly because the relative low frequency of the icebergs and partly due to the fact that the iron limitation is not as severe in the PFZ as the other two zones.

## 3.4 Correlation between annual NPP and small iceberg frequency

Results of local correlation between annual NPP and small iceberg frequency at each grid points are presented in Figure 5 and summarized in Table 3. Only Pearson's $r$ is reported, as Spearman's $rho$ gives similar results. Most significant positive correlations occur in the SIZ, with the median coefficient $r$ at 0.15. The correlation is statistically significant at 0.05 level for 33% of the grids in this zone. The correlation is much weaker for other zones (Table 3). Within each zone, there is a wide range of coefficient values, both positive and negative. Only a small portion of the correlations are statistically significant. This is largely because when summarized at annual timescale, the size of the data is largely reduced. With the data coverage of only 13 years, at each grid point, the number of data is 13 or less, making it difficult to establish a reliable correlation with statistical significance. When data are pooled together, zonal correlation shows much higher statistical strength, indicated by very low $p$ values for the correlation in all zones. Correlation coefficients, however, are still relatively low (Table 3). In general, it seems that correlation is more positive and stronger in the high latitude zones of the SIZ and the CSZ, and weaker in the POOZ and the PFZ.

The general low level of correlation is not surprising for several reasons. First of all, the correlation is only calculated between NPP and small iceberg frequency. The occasional presence of large icebergs could sometimes skew the relationship. However, the relatively low frequency of large icebergs makes it difficult to incorporate them into the same correlation analysis. On the other hand, the low frequency also means that large icebergs could not in general exert a significant influence on the correlation analysis. The direct correlation between NPP and iceberg frequency could also be weakened by other sources of Fe (such as meltwater, sedimentary, and upwelling sources), variable phytoplankton Fe:C quotas, light and silicate limitations, the atmospheric Fe dissolution kinetics, aeolian Fe sources and transport pathways. In addition, the statistical relationship between the SO NPP and icebergs also depends on the forms of Fe in icebergs. Most previous studies focus on direct measurement of soluble iron. However, Hassler and Schoemann (2009) found no direction link between soluble and bioavailable Fe because organic ligands have the differential effects on the solubility and the bioavailability. Several studies seem to suggest that organic colloidal Fe provides an important pool to sustain Fe bioavailability to phytoplankton (e.g. Nodwell and Price, 2001; Chen et al., 2003; Wang and Dei, 2003). Finally, icebergs themselves could have other effects on NPP. For example, Arrigo and van Dijken (2004) observed that icebergs could have negative effects on the polar marine ecosystems in some occasions. Many of these icebergs have long residence times, and hence are likely to have been located in highly productive coastal waters during the peak growing seasons of austral spring and summer. Their presence can alter normal advection patterns of annual sea-ice, and hence the fraction of open water available for phytoplankton growth. All these factors could have contributed to a relatively weak correlation between the SO NPP and iceberg occurrences. For more detailed studies on the impact of icebergs

relative to some of the factors listed above, we need to develop quantifiable indicators for these factors and incorporate them into statistical modelling. Such indicators could include distance to the coast, distance to aeolian dust sources alone transport pathways based on atmospheric circulation patterns, proportions of bioavailable iron to total soluble iron from iceberg case studies, ocean circulation patterns etc.

## 3.4 Spatial distribution of large icebergs and their impact on NPP

Large icebergs are relatively rare. For the period 2002-2014, 393 icebergs were identified in the BYU dataset. Among them, 154 are icebergs larger than 18.5 km in length. They are named icebergs also monitored by the National Ice Centre. The other 239 are smaller icebergs between 5 and 18.5 km. Their tracks are presented in Figure 6a, and the summarized gridded count data in Figure 6b. The majority of large icebergs concentrate near the coastal region of Antarctica, where they are calved from the major ice shelves, such as the Ross, Filchner, Ronne, Larsen, and Amery. Large number of icebergs are also present in the south Atlantic section of the SO, originated mostly from the ice shelves in the Weddell Sea.

Based on the methodology outlined in section 2.4, we calculated the mean NPP for grids occupied by icebergs (iceberg grids), grids immediately adjacent to iceberg grids (adjacent grids), and the rest of the grids in the 7x7 degree window, i.e. grids in the vicinity but further away (nearby grids) at monthly timescale. Results are summarized by ecological zones (Table 4). Most of the iceberg grids are located in the CSZ and the SIZ, but their frequency is still relatively low, at 73 and 60 grids per year respectively. In general, the difference in mean NPP between iceberg grids and adjacent grids are fairly small, but the mean NPP of combined iceberg and adjacent grids are significantly (about 10%) higher than the nearby grids. This pattern is fairly consistent in most of the ecological zones with the only exception of the POOZ, where iceberg/adjacent grids have similar NPP as the nearby grids. Seasonally, the most significant increase of NPP from surrounding grids occurs in the most productive months in the austral summer (December, January and February). Enhanced levels of NPP are often found with both small and large icebergs, although their zonal response is different. With small icebergs, the largest enhancement is seen in zones relatively poor in iron such as the POOZ and the SIZ, whereas the effect in the relatively iron-rich CSZ is not significant. The large icebergs, on the other hand, seem to increase NPP more in the higher latitudes such as the CSZ and the SIZ. However, the low frequency of large icebergs makes it difficult to establish reliable statistical relationship. Moreover, the coarse spatial resolution (getting even coarser at lower latitudes of the POOZ and PFZ) could make it difficult to detect enhancement of NPP near large icebergs.

## 3.5 Temporal trend of icebergs and NPP

Over the period of 1992-2014, the amount of small icebergs has a rather notable increasing trend (Figure 7). Annual iceberg volume increases at 2.6% per year, and iceberg frequency increases at 4.7% per year. Both trends are statistically significant (Table 5). Iceberg amount increases the most for the period 1992-2004, and decrease slightly since then. In particular, there seems to be a period of rapid increase in the first half of the 2000s. Large iceberg frequency, measured as the mean annual number of grids occupied by large icebergs, follows a similar increasing trend (Figure 8) at 7.09% per year (Table 6). Large

iceberg frequency remained relatively low in the 1990s, but experienced a dramatic increase between 2001-2004, and has a slight decreasing trend since 2004. This pattern is quite consistent with the temporal trend observed in the small iceberg frequency. This could be linked to other observed changes in ice discharge and volume loss from Antarctic ice shelves. For example, Mouginot et al. (2014) observed that ice discharge from the Amundsen Sea Embayment increased by 77% from 1973 to 2013, and more than half of the increase occurred between 2003-2009. Paolo et al. (2015) noted that average ice-shelf volume change accelerated from negligible loss at $25 \pm 64$ km$^3$ per year for 1994-2003 to rapid loss of $310 \pm 74$km$^3$ per year for 2003-2012. Our iceberg data seem to confirm a rapid increase in ice discharge in the SO in the early 2000s.

For the period 2002-2014 when NPP data are available, small iceberg volume decreases at a rate of 4.0% per year, and frequency decreases at 1.2% per year, although their mean values are much higher than those for the entire period of 1992-2014. Given the shortness of the period, only iceberg volume trend is statistically significant ($p$=0.01), whereas frequency trend is not. Similarly, large iceberg frequency also decreases slightly at 2.96% per year ($p$=0.02). If icebergs play an important role in enhancing the SO NPP, it is probably not surprising to see the lack of any observable trend in the total NPP of the SO for the data period of 2002-2014 (Figure 9), given the slight negative to no trends in iceberg presence for the same time period. A longer time series is needed to establish whether iceberg and NPP follow a similar temporal trend.

For the period 1992-2014, both volume and frequency of small icebergs increase in almost all ecological zones (Figure 7 and Table 5). The most significant increase occurs in the SIZ (3.5% per year for volume and 6% per year for frequency), which contains over two-thirds of the total iceberg volume. Frequency also increases significantly in the CSZ at 4.28% per year, although iceberg volume shows not trend in this zone. The trend in most of the other zones is relatively small and not statistically significant. For the period of 2002-2014, most zones show decreasing trend for both volume and frequency, except slight increase in frequency in the SIZ. However, most of these trends are not statistically significant, particularly trend in the SIZ, which has the majority of the icebergs. Large iceberg frequency has similar increasing trends in or ecological zones (Table 6) between 6.5 – 8.5% per year for the period 1992-2014. All trends are statistically significant. They have small decreasing trends for 2002-2014, and most of them with no statistical significance.

Many studies suggest that warming has caused mass loss of Antarctica ice sheets, largely through enhanced basal melting and iceberg calving driven by warming of the ocean (Rignot et al., 2013; Liu et al., 2015). Iceberg calving is a major contributor of mass loss in Antarctica ice shelves, accounting for 1/3 (Liu et al., 2015) to 45% (Rignot et al., 2013) of the total mass loss. However, the mass change is not spatially uniform (Shepherd et al., 2012; Hanna et al., 2013; Sasgen et al., 2013). West Antarctica and the Antarctic Peninsula have experienced the largest mass loss with observations suggesting enhanced basal melting (Jacobs et al., 2011; Pritchard et al., 2012; Rignot et al., 2013), accelerated glacial flows (Pritchard et al., 2009) and fast retreat of grounding lines (Jenkins et al., 2010; Rignot et al., 2014). On the other hand, the ice sheet on East Antarctica (facing the Atlantic and Indian Ocean) is relatively stable (Shepherd et al., 2012). Some thickening and mass gain has been observed for much of the coastal region of the East Antarctica Ice sheet except a small section near Victoria Land (135$^o$ E to 165$^o$ E) facing the Indian Ocean, where slight decrease in ice mass is recorded (Shepherd et al., 2012). In order to examine if iceberg quantity has responded to such changes, we examine the three sections of the SO separately, and the results for small

icebergs are presented in Figure 10 and Table 7. For the period 1992-2014, the SP section, adjacent to West Antarctica, shows large increase in iceberg quantity. The total volume is increasing at 4.6% per year, and frequency at 6.8% per year. Both trends are highly significant statistically. SI section shows similar trend of increases. However, in the SA section the iceberg amount is relatively unchanged. There is a slight decrease in iceberg volume, and a slight increase in iceberg frequency. However, neither trend is statistically significant. Over the period 2002-2014, the SA section experiences significant decrease in both iceberg volume (12.6% per year) and frequency (10.4% per year). The SP section shows a significant increase in frequency (5.4% per year), but the increase in total volume (4.0% per year) is not statistically significant. The SI section shows a slight increase in both volume and frequency, but neither trend is statistically significant. Such differentiated temporal trend in icebergs at different sections of the SO could be related to the spatial variation of mass change in the Antarctica ice sheets, which could in turn be linked to climate change in Antarctica. West Antarctica is among the most rapidly warming regions on Earth (Bromwich et al., 2013). Despite an overall sea surface temperature (SST) cooling around Antarctica, warming trends have been observed for the southeast Indian Ocean sector, in the Weddell, Bellingshausen and Amundsen Seas of West Antarctica (Jones et al., 2016). Results for large icebergs are presented in Fig 11 and Table 8. They seem to present a somewhat different picture. Large iceberg frequency is increasing in both S. Atlantic (7% per year) and S. Indian (9.3% per year) Ocean, but remains relatively unchanged in the S. Pacific apart from a small increase in the 2000s (Figure 11). Large icebergs have very uneven spatial distribution, with 57% occurring in the S. Atlantic, 34% in the S. Indian, and only 9% in the S. Pacific Ocean. Many large icebergs have originated from the Ronne-Filchner Ice Shelf bordering the Weddell Sea, which drift into the S. Atlantic part of the SO, whereas relatively few large icebergs have originated from ice shelves in the Western Antarctica, which drift into the S. Pacific section of the SO. This overall low frequency may have contributed to the lack of temporal trend in the number of large icebergs in this area despite larger mass loss of ice shelves in the Western Antarctica. However the reason for the disparity in the sectional trends between small and large icebergs remains unclear, and needs further investigation.

# 4 Conclusions

This study aims to examine whether icebergs have a significant impact on the ocean NPP at the scale of the SO. Through using remote sensing data, we examine the impacts of both small and large icebergs on the ocean NPP. We divided the SO into four ecological zones based on their different nutrient source and profile. For small icebergs, we compared NPP for grids with and without iceberg presence within each zone. We found that in many places grids with iceberg presence have higher NPP than those without icebergs. However, the impact is not uniform. In the CSZ where high level of iron is supplied through glacial meltwater and sediment input from the continent and continental shelf and phytoplankton growth is largely limited by macronutrients, the presence of icebergs does not seem to have any impact on the ocean NPP. On the other hand, Iceberg presence is associated with significantly higher NPP values in the HNLC regions. The NPP of grids with icebergs is 21% higher than those without in the SIZ, and 16% higher in the POOZ. Direct correlation between iceberg frequency and NPP is

weak although statistically significant. Strongest correlation is found at the SIZ, which contains over 70% of the icebergs by volume. For large icebergs, we examine the average NPP of iceberg occupied grid cells, immediately adjacent cells, and nearby cells that are further away. We found that NPP of iceberg cells and adjacent cells is on average 10% higher than NPP of nearby cells. The positive impact of large icebergs is stronger in high latitude zones of the CSZ and the SIZ, where most of them occur. Therefore, icebergs, large and small, have an observable positive association with ocean NPP at the SO scale. For the entire period of the iceberg data, 1992-2014, both large and small icebergs have shown significant increasing trends. The increase is most rapid during the first half of 2000s, and levels off since then. For small icebergs, the increasing trend is most notable for the Pacific and Indian sections of the SO, whereas the Atlantic section of the SO shows no statistically significant trend. This could be related to the greater mass loss of the West Antarctica Ice Shelf, and the relative stability of the East Antarctica Ice Shelf under present climate change. The sectional trends are different for large icebergs, which increase significantly for the S. Atlantic and S. Indian sections, but remain relatively unchanged in the S. Pacific section of the SO. The very low frequency of icebergs in the S. Pacific section of the SO makes it harder to detect any trend. However, the exact mechanism that accounts for the difference in sectional trends between small and large icebergs remains unclear. As the climate continues to warm, the Antarctic Ice Sheet is expected to experience increased mass loss as a whole, which could lead to more icebergs in the region. Based on the positive association between icebergs and NPP shown in this study, this could result in higher level of NPP in the SO as a whole, providing a possible negative feedback for global warming in the near future.

**Acknowledgement**

The research was supported by the State Oceanic Administration (CHINARE2012-02-02) and the Natural Science Foundation of China (41330526, and 41571180). We would like to thank Dr. Jean Tournadre from the Laboratoire d'Océanographie Spatiale for his kind assistance with the Altiberg small iceberg dataset, and Dr. Shenfu Dong from the Scripps Institution of Oceanography for providing the polar front location data.

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

**Tables**

Table 1. Mean annual primary production of ecological zones of the SO 2002-2014.

| Zones | Total Area (million km²) | Area of productive grids (million km²) | Total Annual Production (Gt C) | NPP over productive waters (gC m⁻² yr⁻¹) | NPP over all waters (gC m⁻² yr⁻¹) |
|---|---|---|---|---|---|
| CSZ | 3.10 | 0.54±0.01 | 0.06±0.003 | 113.5±3.7 | 19.6±1.1 |
| SIZ | 16.54 | 6.41±0.08 | 0.40±0.010 | 63.0±1.6 | 24.1±0.6 |
| POOZ | 14.44 | 10.87±0.02 | 0.63±0.010 | 57.5±1.0 | 43.6±0.7 |
| PFZ | 6.12 | 4.97±0.01 | 0.38±0.004 | 76.5±0.8 | 62.7±0.7 |

Table 2. Mean annual NPP for grids with and without icebergs 2002-2014.

| Zone | Total area of grids with icebergs (million km²) | Area of Productive grids with icebergs (million km²) | Annual iceberg volume (Gt) | Annual iceberg volumes over productive water (Gt) | NPP over productive waters with icebergs (gC m⁻² yr⁻¹) | NPP over productive waters without icebergs (gC m⁻² yr⁻¹) | Difference in NPP of cells with and without icebergs (gC m⁻² yr⁻¹) |
|---|---|---|---|---|---|---|---|
| CSZ | 0.20±0.01 | 0.17±0.01 | 104±12.5 | 89±9.8 | 113±7.0 | 114±4.9 | -0.9 |
| SIZ | 2.66±0.07 | 2.34±0.07 | 1351±78.8 | 1258±65.4 | 70±1.4 | 58±1.7 | 11.8* |
| POOZ | 1.33±0.14 | 1.10±0.11 | 371±60.3 | 315±59.2 | 67±1.6 | 57±1.1 | 10.3* |
| PFZ | 0.26±0.05 | 0.23±0.04 | 64±14.1 | 55±12.3 | 83±2.4 | 75±0.8 | 8.1* |

* Statistically significant at 0.01 level based on Student's t test.

Table 3. Correlation between annual total primary production and iceberg frequency.

| Zone | Min. | 1st Quarter | Median | Mean | 3rd Quarter | Max. | % of grids significant** | Pearson's r | Spearman's rho |
|---|---|---|---|---|---|---|---|---|---|
| | | | Local correlation (Pearson's r) | | | | | Global correlation | |
| CSZ | -0.40 | -0.06 | 0.10 | 0.07 | 0.20 | 0.66 | 16.30 | 0.16* | 0.23* |
| SIZ | -0.55 | 0.04 | 0.15 | 0.14 | 0.25 | 0.57 | 33.49 | 0.18* | 0.22* |
| POOZ | -0.36 | -0.04 | 0.06 | 0.04 | 0.13 | 0.43 | 21.54 | 0.13* | 0.16* |
| PFZ | -0.27 | -0.10 | 0.00 | -0.01 | 0.09 | 0.23 | 14.50 | 0.12* | 0.14* |

Note: * Correlation significant at 0.01 level based on Student's t test.

** % of grids with iceberg presence for which correlation is significant at 0.05 level based on Student's t test.

Table 4. Impact of large icebergs on NPP: comparison of mean NPP of iceberg grids, adjacent grids and nearby grids.

| Zone | Mean NPP of iceberg grids (mgC m$^{-2}$ d$^{-1}$) | Mean NPP of adjacent grids (mgC m$^{-2}$ d$^{-1}$) | Mean Diff. | p* value | Mean NPP of iceberg and adjacent Grids (mgC m$^{-2}$ d$^{-1}$) | Mean NPP of nearby grids (mgC m$^{-2}$ d$^{-1}$) | Mean diff. | p* value | No. of Iceberg grids per year |
|---|---|---|---|---|---|---|---|---|---|
| CSZ | 241±6.5 | 241±5.9 | 0.62 | 0.84 | 242±6.0 | 215±4.6 | 27.17 | 0.00 | 73.08 |
| SIZ | 206±5.0 | 210±5.1 | -4.61 | 0.05 | 209±4.9 | 189±3.8 | 20.03 | 0.00 | 60.92 |
| POOZ | 151±5.8 | 148±5.1 | 2.19 | 0.25 | 149±5.2 | 149±5.0 | 0.13 | 0.94 | 14.23 |
| PFZ | 249±15.5 | 257±14.8 | -8.91 | 0.02 | 256±15.0 | 243±12.7 | 13.06 | 0.02 | 3.77 |
| Total | 219±3.7 | 220±3.5 | -1.72 | 0.33 | 220±3.5 | 199±2.8 | 21.31 | 0.00 | 152 |

* p values are calculated based on the pair-wise t test.

Table 5. Temporal trend of annual volume and frequency of small icebergs by ecological zones.

| Time | Zone | Iceberg Volume | | | | Normalized Iceberg frequency | | | |
|---|---|---|---|---|---|---|---|---|---|
| | | Annual Mean (Gt) | Rate of Change (Gt year$^{-1}$) | % change | p* value | Annual Mean (10$^{-4}$) | Rate of Change (10$^{-4}$ year$^{-1}$) | % change | p* value |
| 1992-2014 | CSZ | 100±8.3 | -0.98 | -0.98 | 0.47 | 4.49±0.40 | 0.19 | 4.28 | 0.00 |
| | SIZ | 1099±79.6 | 38.21 | 3.48 | 0.00 | 9.28±0.89 | 0.55 | 5.96 | 0.00 |
| | POOZ | 333±40.3 | 1.70 | 0.51 | 0.80 | 4.19±0.45 | 0.03 | 0.73 | 0.67 |
| | PFZ | 49±10.0 | 1.42 | 2.88 | 0.38 | 1.84±0.24 | -0.01 | -0.29 | 0.89 |
| | All | 1581±107 | 40.35 | 2.55 | 0.01 | 6.42±0.52 | 0.30 | 4.65 | 0.00 |
| 2002-2014 | CSZ | 104±12.5 | -9.97 | -9.61 | 0.00 | 5.74±0.43 | -0.08 | -1.42 | 0.54 |
| | SIZ | 1351±78.8 | -24.76 | -1.83 | 0.30 | 12.52±0.64 | 0.12 | 0.95 | 0.55 |
| | POOZ | 371±60.3 | -34.15 | -9.21 | 0.04 | 4.68±0.71 | -0.44 | -9.35 | 0.03 |
| | PFZ | 64±14.1 | -6.64 | -10.43 | 0.11 | 1.95±0.41 | -0.23 | -11.91 | 0.04 |
| | All | 1889±105.8 | -75.52 | -4.00 | 0.01 | 8.29±0.35 | 0.30 | -1.24 | 0.33 |

5    *p value is calculated based on the F test.

Table 6. Temporal trend of large icebergs: mean annual number of grids occupied by large icebergs by ecological zones.

| Time | Zone | Mean annual Iceberg grids | Rate of change per year | % change per year | *p*-value* |
|---|---|---|---|---|---|
| 1992-2014 | CSZ | 51±6.6 | 3.46 | 6.80 | 0.00 |
| | SIZ | 40±5.5 | 2.86 | 7.20 | 0.00 |
| | POOZ | 10±2.5 | 0.79 | 7.85 | 0.03 |
| | PFZ | 2±0.5 | 0.17 | 8.58 | 0.03 |
| | All | 103±13.9 | 7.28 | 7.09 | 0.00 |
| 2002-2014 | CSZ | 76±9.1 | -2.02 | -2.66 | 0.04 |
| | SIZ | 60±7.6 | -1.14 | -1.89 | 0.30 |
| | POOZ | 17±3.4 | -1.16 | -6.86 | 0.21 |
| | PFZ | 4±0.7 | -0.32 | -8.55 | 0.08 |
| | All | 157±19.1 | -4.63 | -2.96 | 0.02 |

*p value is calculated based on the *F* test.

Table 7. Temporal trend of annual volume and frequency of small icebergs by sections of the SO.

| | | Iceberg Volume | | | | Normalized Iceberg frequency | | | |
|---|---|---|---|---|---|---|---|---|---|
| Time | Section | Annual Mean (Gt) | Rate of Change (Gt year$^{-1}$) | % change | *p** value | Annual Mean (10$^{-4}$) | Rate of Change (10$^{-4}$ year$^{-1}$) | % change | *p** value |
| 1992-2014 | SA | 751±87.9 | -1.18 | -0.16 | 0.93 | 4.73±0.54 | 0.09 | 1.81 | 0.32 |
| | SI | 393±41.0 | 21.08 | 5.37 | 0.00 | 2.51±0.21 | 0.11 | 4.35 | 0.00 |
| | SP | 490±47.6 | 22.33 | 4.56 | 0.00 | 3.48±0.35 | 0.24 | 6.75 | 0.00 |
| | All | 1634±110.4 | 42.23 | 2.58 | 0.01 | 3.57±0.28 | 0.16 | 4.38 | 0.00 |
| 2002-2014 | SA | 823±126.0 | -103.96 | -12.64 | 0.00 | 5.68±0.76 | -0.59 | -10.35 | 0.00 |
| | SI | 524±47.5 | 7.56 | 1.44 | 0.61 | 4.73±0.26 | 0.09 | 1.31 | 0.61 |
| | SP | 612±61.6 | 24.24 | 3.96 | 0.19 | 2.51±0.34 | 0.11 | 5.40 | 0.00 |
| | All | 1958±106.5 | -72.16 | -3.69 | 0.01 | 3.48±0.18 | 0.24 | -1.15 | 0.34 |

5    *p value is calculated based on the *F* test.

Table 8. Temporal trend of number of grids occupied by large icebergs by sections of the SO.

| Time Period | section | Mean Annual Iceberg Grids | Rate of Change | % change per year | p-value* |
|---|---|---|---|---|---|
| 1992-2014 | SA | 61±8.5 | 4.27 | 6.99 | 0.00 |
| | SI | 35±5.5 | 3.29 | 9.30 | 0.00 |
| | SP | 10±2.5 | 0.24 | 2.36 | 0.55 |
| | All | 106±14.6 | 7.80 | 7.32 | 0.00 |
| 2002-2014 | SA | 92±7.0 | -2.44 | -2.66 | 0.19 |
| | SI | 58±3.4 | 0.98 | 1.70 | 0.29 |
| | SP | 14±3.6 | -2.71 | -19.30 | 0.00 |
| | All | 163±8.2 | -4.17 | -2.56 | 0.04 |

*p value is calculated based on the *F* test.

**Figures**

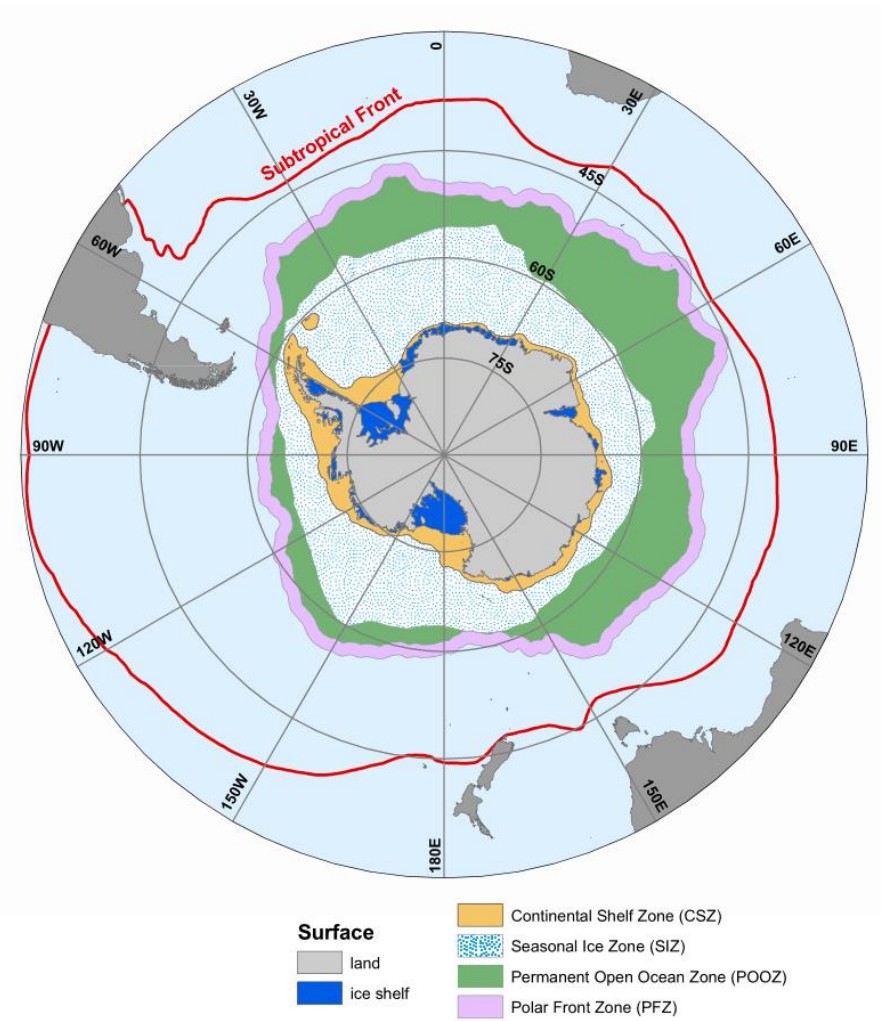

Figure 1. Ecological zones of the Southern Ocean.

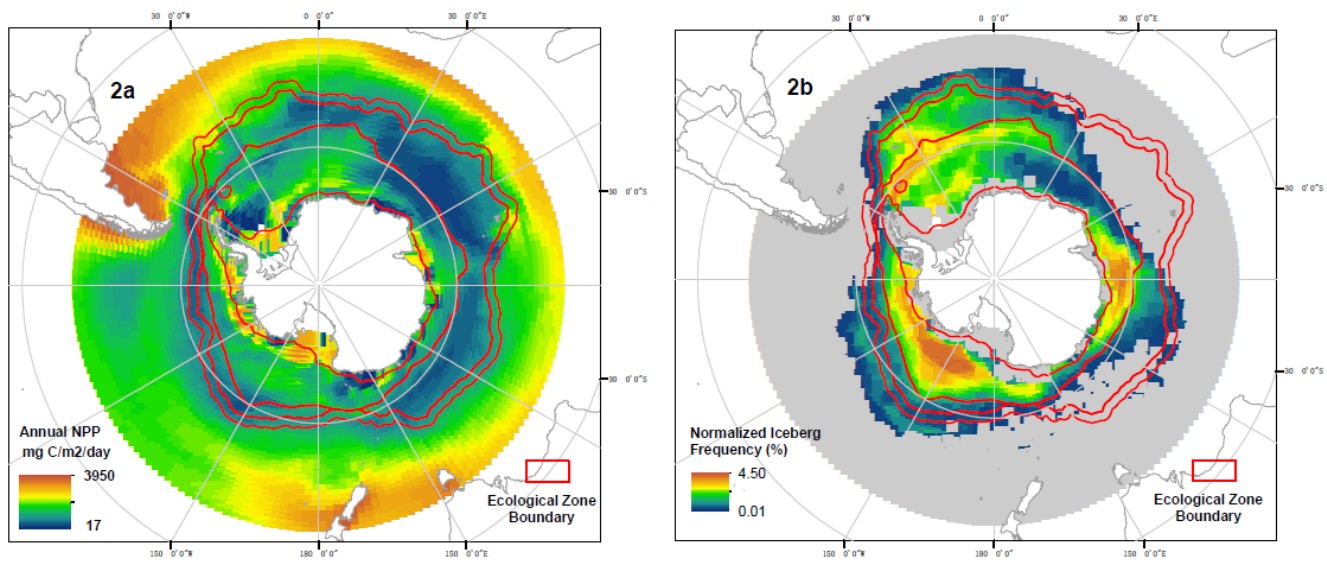

Figure 2. Spatial distribution of annual mean NPP (1a) and normalized iceberg frequency (1b) in relation to ecological zones.

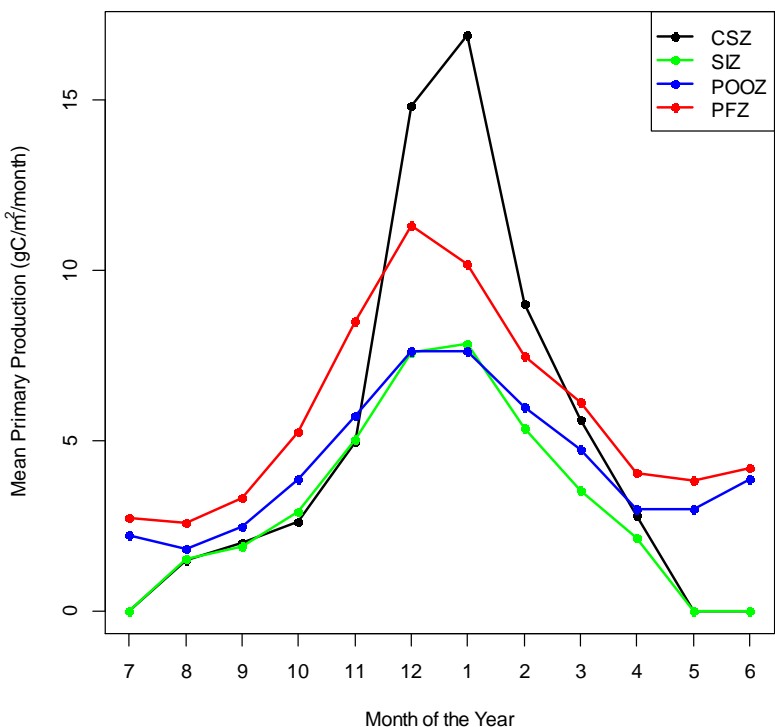

Figure 3. Monthly mean NPP of different ecological zones 2002-2014.

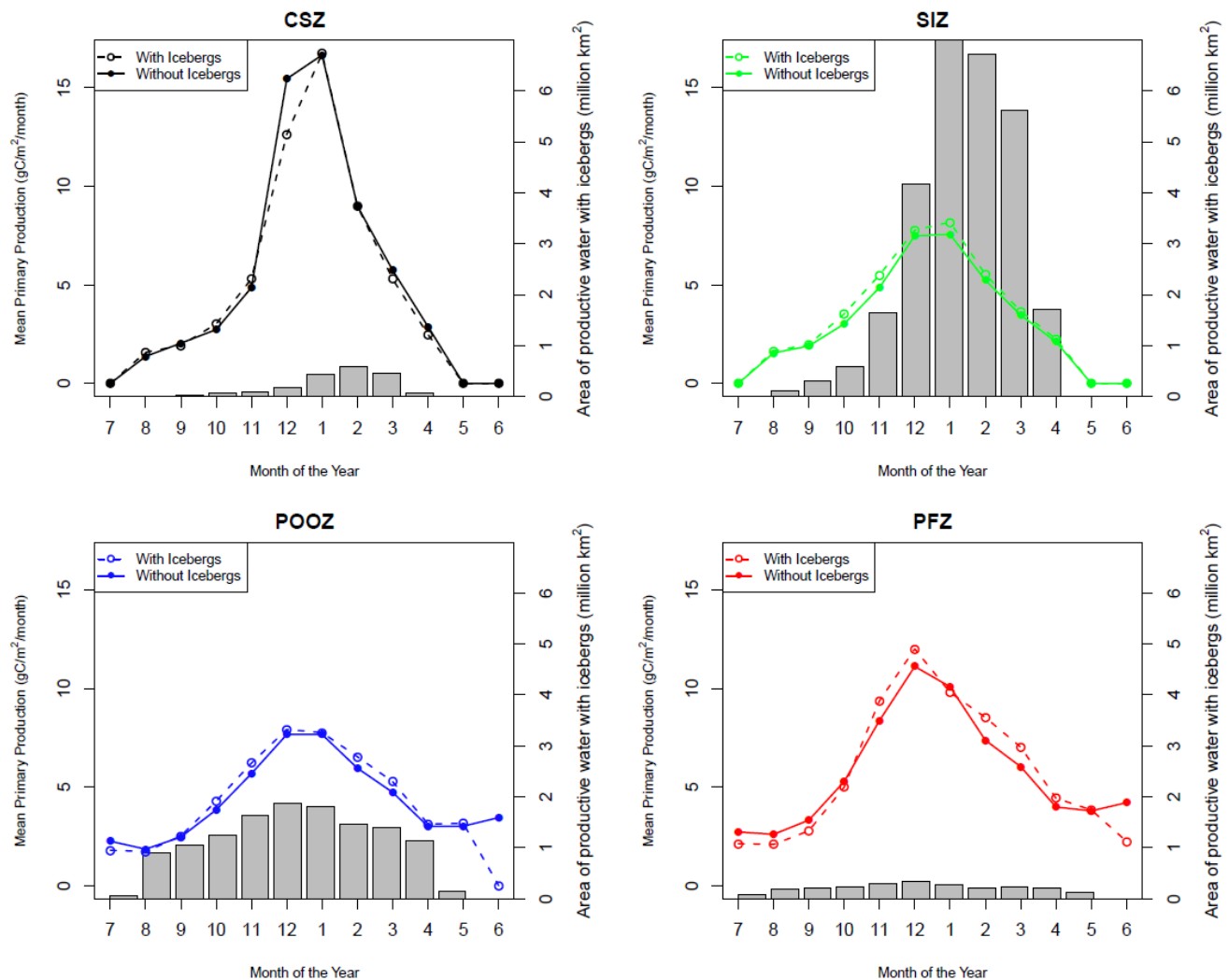

Figure 4. Monthly NPP for grid cells with (solid line) and without icebergs (dashed line) for each zone. Bar charts indicate the area of productive grids with icebergs.

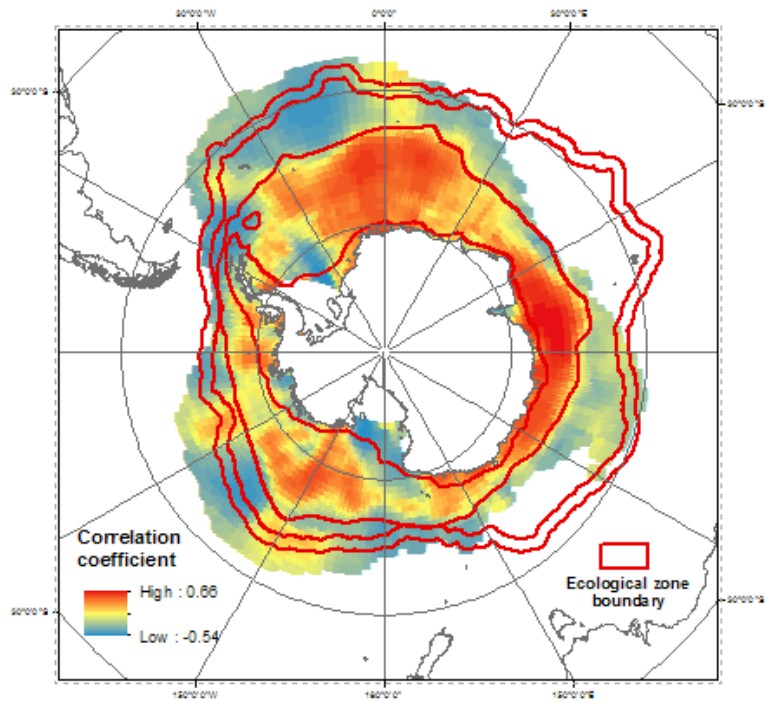

Figure 5. Temporal correlation between annual NPP and normalized iceberg frequency.

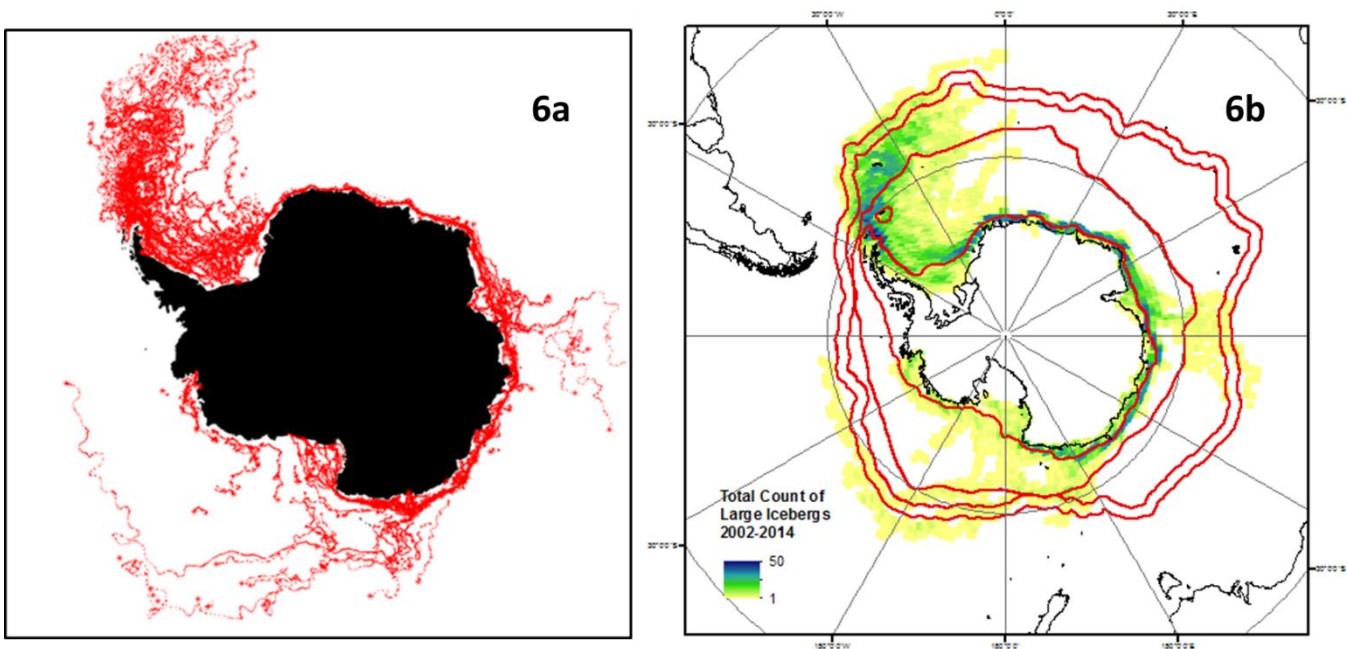

Figure 6. Spatial distribution of large icebergs. 6a indicates all iceberg tracks between 1999 and 2010 (from
http://www.scp.byu.edu/data/iceberg/database1.html). 6b shows the gridded count of iceberg occurrences between 2002 and 2014.

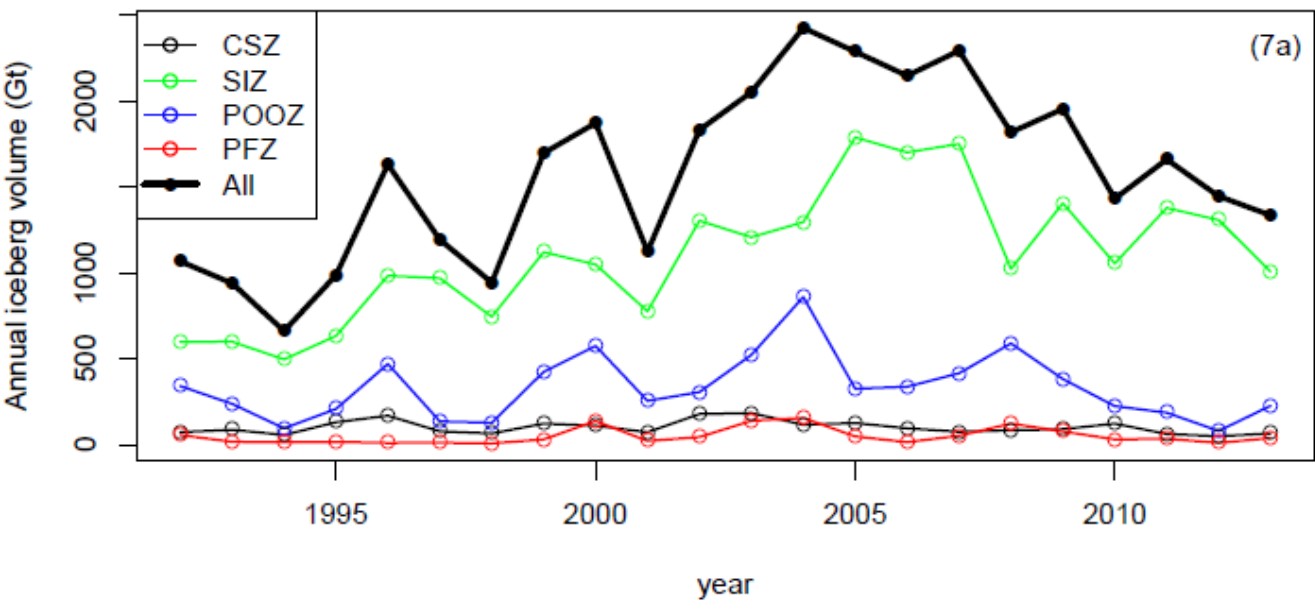

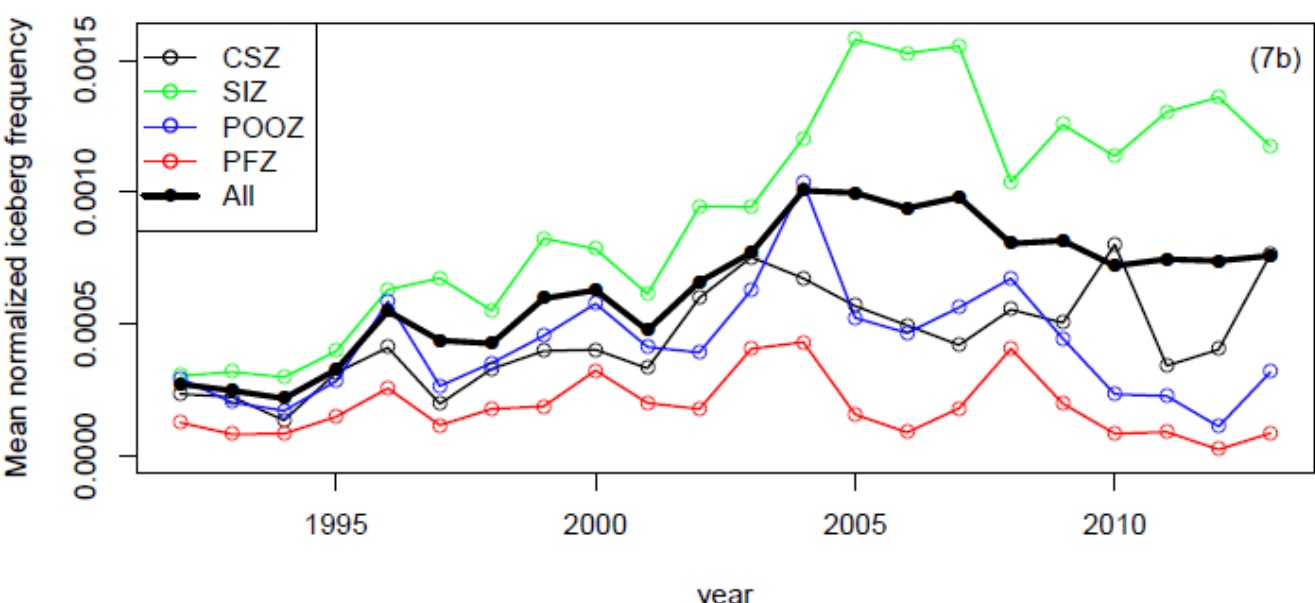

Figure 7. Temporal trend of annual total volume (6a) and mean frequency (6b) of small icebergs by ecological zones of the SO.

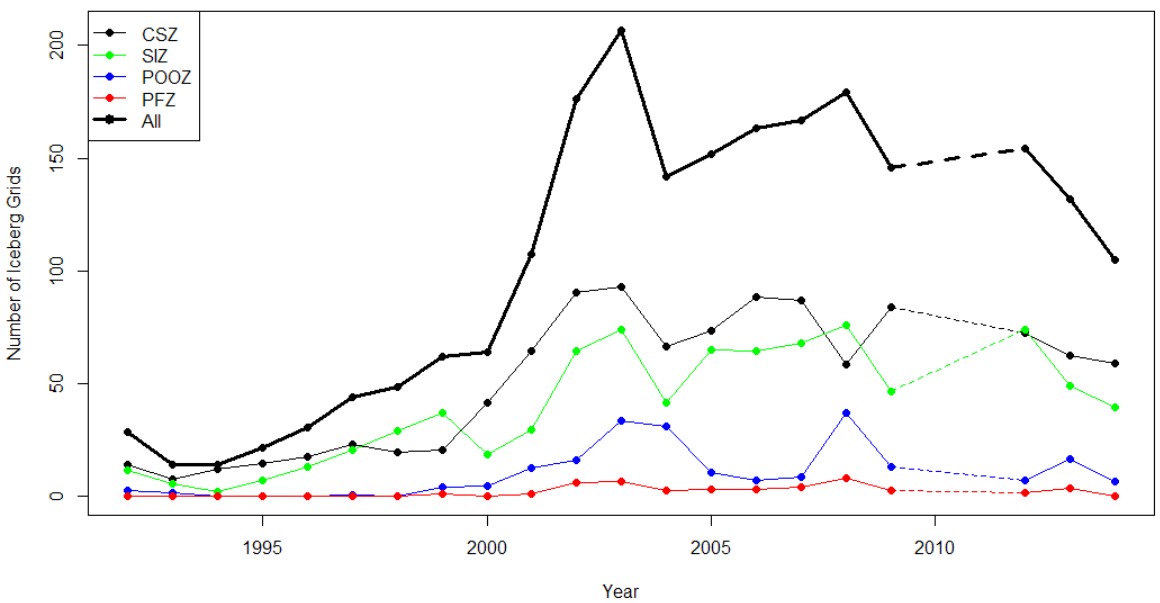

Figure 8. Temporal trend of annual number of grids occupied by large icebergs 1992-2014.

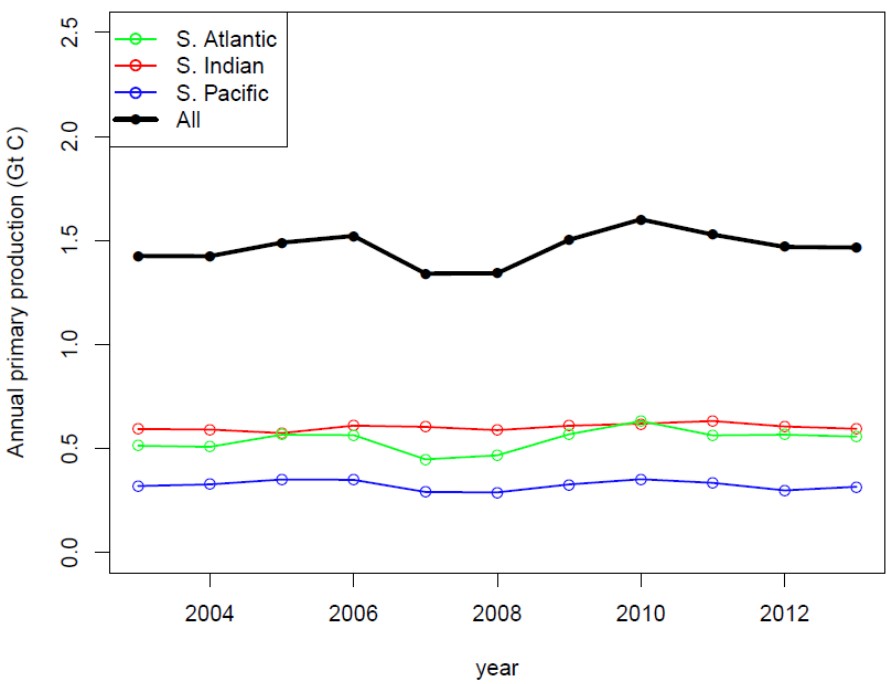

Figure 9. Temporal trend of total annual primary production by sections in the SO.

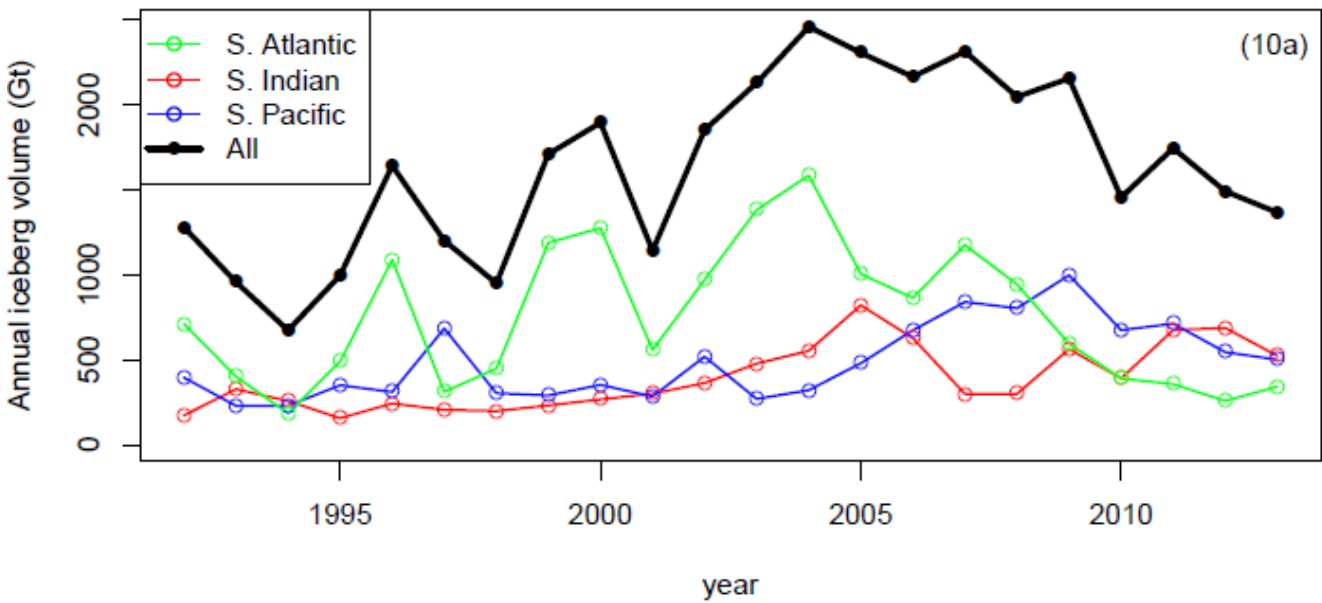

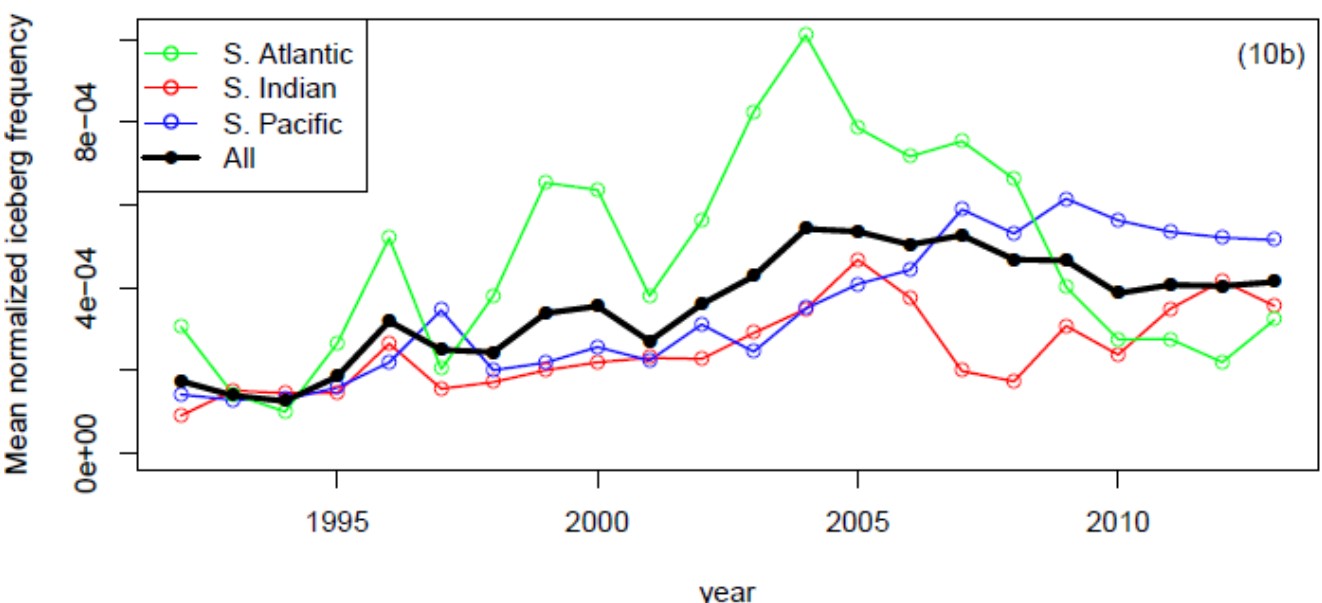

Figure 10. Temporal trend of annual total volume (10a) and mean normalized frequency (10b) of small icebergs by sections of the SO 1992-2014.

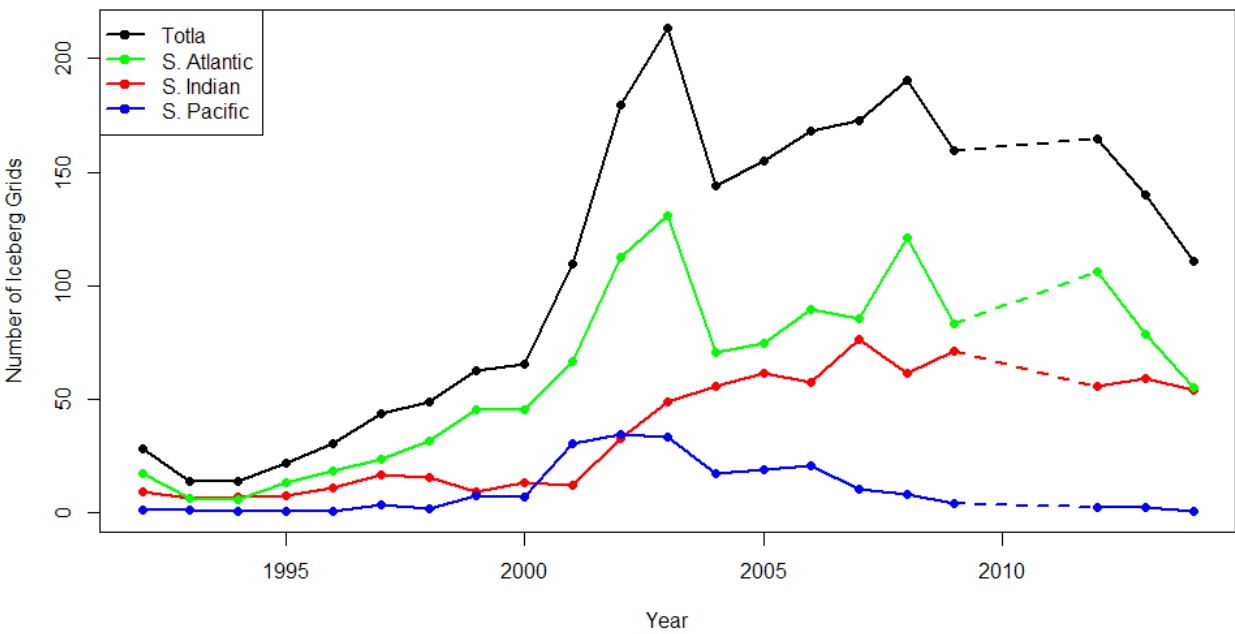

Figure 11. Temporal trend large icebergs: mean annual number grids occupied by large iceberg 1992-2014.