# Peer review of "Impact of Icebergs on Net Primary Productivity in the Southern Ocean"

_The Cryosphere, 2016_

## Referee Comment (RC1) · Anonymous Referee #1 · 24 Aug 2016

The authors address the interesting idea that icebergs may play a role in stimulating primary production in the Southern Ocean, through supply of bioavailable iron to nutrient limited areas. Their approach is to use two remote sensing datasets (one for ocean productivity and another for iceberg probability), and through multiple linear regression analysis to examine the effect of temperature and iceberg concentration on net primary productivity. While I agree that this issue is one that deserves attention, I have significant concerns with many of the assumptions that underpin the analysis, and in the end feel that the authors conclusions are not supported by the work presented here. Below I detail my points for the authors to consider:

1) There seems to be a rather large disconnect in scale here that I'm having trouble understanding. It appears that most of the previous work on this subject has looked at iceberg properties and associated ocean/productivity effects at small scales – i.e.,

individual icebergs or icebergs within small defined areas. How do the authors justify scaling up to a hemispheric-scale analysis based on these limited findings? One could imagine that icebergs from different regions would have radically different properties – volume, area, number, sediment concentration, sediment composition, etc. In particular, the bioavailable iron concentration would almost surely be different across various regions – how does one then compensate for that when scaling to the entire Southern Ocean?

2) The most recent work on the subject (Duprat et al., 2016) focused on large icebergs and their potential ocean impact. The authors make the assumption (lines 30-35) that the majority of small icebergs are associated with large ones, and therefore that total iceberg concentration should reveal similar impacts on ocean productivity. It's not clear to me what the basis is for this assumption. What is all or most of the sediment and hence available iron is limited to the large bergs? Smaller ones may simply be fragments that were never in contact with the bed surface. I'll readily admit here that I'm not an expert on iceberg processes, but this assumption is fundamental to the authors approach and it just seems unfounded to me.

3) The remote sensing product used to estimate iceberg probability is described briefly, but again I am struggling to understand the connection between iceberg probability (presumably a derived product and not a direct measurement) and the relevant information in this case – namely the amount of iron going into the water. Moreover, a broad-scale gridded remote sensing product seems rather coarse to evaluate small-scale processes. Again, I am not a remote sensing expert, and perhaps the Tournadre et al. 2012 product is a gold standard in the field, but its application here may be somewhat misleading.

4) My concerns about scaling and data products aside, the results of the statistical analyses are just not that compelling. There is a positive correlation between NPP and iceberg probability (not presence or concentration, as the authors write), but in all the cases the variance explained by iceberg probability is so small it's hard to imagine

any physical significance. And in particular, the variance explained by temperature vs. icebergs makes the iceberg effect quite small. What of other variables? Ocean productivity could be equally linked to any number of other factors (e.g., wind stress, upwelling, circulation, etc.) – why did the authors choose just temperature (and icebergs) for the MLR analysis? And of course, old adage that correlation does not prove causation applies here – even with a significant correlation between icebergs and NPP (albeit with ~2% of variance explained), the authors provide no argument for the linkage and a SO scale.

5) The discussion section provides a useful summary of all the reasons why there may be such a small statistical effect of icebergs on productivity, and indeed the authors cite studies suggesting that the presence of icebergs may in fact decrease local productivity. In light of all of this, I find the final 5 lines of the conclusion section to be unsupportable by the existing analysis and results. Overall I think the authors need to either a) include a significant number of additional analyses to bolster their arguments, or b) carefully consider what conclusions can actually be drawn from the existing analysis, and re-write the paper and title accordingly.

---

## Referee Comment (RC2) · Anonymous Referee #2 · 29 Aug 2016

Wu & Hou use net primary production (NPP; estimated from satellite observations), temperature, and iceberg occurrence frequency data to investigate the variation of NPP with temperature and/or iceberg frequency in the Southern Ocean. They apply multi-linear regression (MLR) where the logarithm of NPP is considered as response to temperature and iceberg frequency; the logarithm of NPP is used to obtain residuals that better fulfil requirements for linear regression. The MLR and calculation of various coefficients of determination ($R^2$) is described in more detail than usual in the current literature showing the carefulness in application of these powerful methods. The authors claim that they found a small, however, statistically significant influence of icebergs.

The correlation coefficients (r) between the relative frequency of icebergs and (logarithm of) NPP and the corresponding coefficients of determination ($R^2$) based on MLR

are small and would speak against an influence of icebergs on NPP. However, this negative result may be, at least in part, due to an inappropriate approach.

By considering the whole region south of 40°S, the authors include large areas that never see an iceberg and where thus icebergs cannot really influence NPP. On the other hand, temperature correlates with almost everything and is, at least on the large scales considered here, also a proxy for latitude, light and maybe other quantities. I'm convinced that large icebergs can have an influence on NPP, however, the question is which mechanisms are at work here (iron supply, upwelling of freshwater, increased mixing) and, depending on the mechanism, how large is the area of influence.

I suggest that the authors look at their results with open mind and discuss limitations of their approach in the light of known as well as speculative mechanisms.

General comments:

The terms 'iceberg probability of presence' and 'iceberg presence probability' should be avoided. I would prefer 'relative frequency of icebergs'.

The interpretation of correlation coefficients depends very much on the context (for example, high-quality measurements in branches of physics versus ecological observations with small sample sizes). A rule of thumb might be 'no or weak correlation' for $-0.3 < r < +0.3$, 'positive correlation' for $r > 0.3$. What's your interpretation of correlation coefficients? What is meant by 'significant' in this context?

'Correlation analysis shows that for all grid cells, NPP is significantly correlated with temperature ($r = 0.66$), but not with iceberg probability ($r = -0.03$). However, if only the cells with iceberg presence are considered, NPP becomes significantly correlated with iceberg probability ($r = 0.12$), whereas the correlation between NPP and temperature greatly weakens ($r = 0.27$) albeit still significant. When temperature is controlled, the correlation between NPP and iceberg probability increases significantly both in case of all grid cells and for cells with iceberg presence. In all cases, NPP is positively

correlated with iceberg probability, suggesting that the presence of iceberg tends to increase NPP in those places.'

When considering the whole oceanic area south of 40°S, many 1° x 1° cells have very low relative frequencies of icebergs and any variation of NPP in these 'low frequency' cell cannot be 'explained' by icebergs. Thus it is not surprising that correlation with iceberg frequency is low (r = -0.03). If restricting the area to cells where the relative frequency of icebergs is larger than zero (is zero really the threshold value), it is not surprising that the value of the correlation coefficient changes, however, r = 0.12 is still very small (-> r^2 = 0.01!!!). I don't understand what is meant by 'when temperature is controlled'. A correlation coefficient of 0.27 is in my opinion a borderline case.

p.6 'However, the effect of the iceberg probability on NPP increases as measured by both R2 (0.02) and standardized coefficient (0.15). This effect is statistically significant at critical level of 0.01.' I don't understand what the authors would like to convey here. R2 = 0.02 is small and thus iceberg frequency is not a good quantity for predicting or explaining variations in NPP. I do not know how the authors calculated a p-value below 0.01 and what it means in the current context. The conclusion is mainly based on these numbers (R2 = 0.02, p < 0.01):

'. . .our analyses show that iceberg presence has a small, yet statistically significant, positive impact on the SO NPP. . . . in places with iceberg presence, iceberg probability could independently explain 2% of the NPP'.

Fig.2: I doubt that comparison of zonally averaged NPP & relative frequency of icebergs yields much insight (please drop figure).

The text needs a bit polishing by a native English speaker (examples: 'planktons', 'which is much contrasted in the three ocean basins')

Specific comments:

- Southern Ocean is defined in the manuscript as the oceanic region south of 40°S

(which is fine with me): you don't have to repeat this definition several times

- abstract: 'NPP in the SO is largely influenced by temperature ...' I suggest reformulation because MLR only shows variation of NPP with temperature and not (direct) 'influence'. Temperature is co-varying with many other quantities in the SO and thus it is not clear by what mechanism NPP is 'influenced' by temperature.

- page1, line 27: planktons -> plankton

- page1, line 30: 'in either natural or artificial settings' you might cite here: Blain, S., Quéguiner, B., Armand, L., Belviso, S., Bombled, B., Bopp, L., ... & Christaki, U. (2007). Effect of natural iron fertilization on carbon sequestration in the Southern Ocean. Nature, 446(7139), 1070-1074.

- Smetacek, V., C. Klaas, V.H. Strass, P. Assmy, M. Montresor, B. Cisewski, N. Savoye, A. Webb, J.M. Arrieta, U. Bathmann, R. Bellerby, G.M. Berg, P. Croot, F. d'Ovidio, S. Gonzalez, J. Henjes, G.J. Herndl, L.J. Hoffmann, H. Leach, M. Losch, M.M. Mills, C. Neill, I. Peeken, R. Röttgers, O. Sachs, E. Sauter, M.M. Schmidt, J. Schwarz, A. Terbrüggen, & D. Wolf-Gladrow, Deep carbon export from a Southern Ocean iron-fertilized plankton bloom, Nature, 487, 313-319, 2012. doi:10.1038/nature11229

p.2 total dissolved Fe in SO: you cite more recent work: Klunder, M. B., Laan, P., Middag, R., De Baar, H. J. W., & Van Ooijen, J. C. (2011). Dissolved iron in the Southern Ocean (Atlantic sector). Deep Sea Research Part II: Topical Studies in Oceanography, 58(25), 2678-2694.

Klunder, M. B., Laan, P., De Baar, H. J. W., Middag, R., Neven, I., & Van Ooijen, J. (2014). Dissolved Fe across the Weddell Sea and Drake Passage: impact of DFe on nutrient uptake. Biogeosciences, 11(3), 651-669.

p.2, lines 9-11: Fe from sediments has to be mixed up or upwelled; Fe source from hydrothermal vents is missing

German, C. R., Legendre, L. L., Sander, S. G., Niquil, N., Luther, G. W., Bharati, L.,

... & Le Bris, N. (2015). Hydrothermal Fe cycling and deep ocean organic carbon scavenging: Model-based evidence for significant POC supply to seafloor sediments. Earth and Planetary Science Letters, 419, 143-153.

p.2, line 16: drop 'Thus'

p.2 'Raiswell and Canfield (2012) recently even suggested that icebergs could supply more than 90% of total colloidal and filterable Fe in the SO.' Raiswell and Canfield (2012) write: 'The model indicates that the rate of delivery of bioavailable Fe from icebergs to the Southern Ocean is at least as large as that by wind-blown dust. However estimates of all the main aqueous, nanoparticulate and colloidal (and potentially bioavailable) Fe inputs to the ocean are poorly-constrained.'

p.4, lines 8-9 '$\mu$ is the mean of the variable and $\sigma$ is the standard deviation of the variable' -> 'm is the mean and s is the standard deviation of the sample' & change eq. accordingly, i.e. z = (x − m)/s

p. 4, lines 26-27 'NPP is relatively high near the coast of Antarctica, largely because of nutrient input from the continent.' I suggest replacing 'nutrient' by 'iron'.

p.4 -60 °S -> 60°S [drop minus sign: S already indicates 'negative' latitudes; no space between degree symbol and S]; please change everywhere in manuscript

---

## Editor Comment (EC1) · E. Brook (Editor) · 13 Sep 2016

Dear Drs. Wu and Hou:

Now that the review phase of this paper is completed I would like to encourage you to submit a response to the reviews for editorial consideration before undertaking a revised manuscript. The reviewers have raised some significant concerns about some of the data analysis and it would be best to sort out the importance of these comments for the conclusions before going forward with a revised manuscript.

With best wishes, Ed

---

## Author Response (AR1)

Dear Editor,

We would like to thank you and the reviewers for their helpful comments. Based on their comments, we have significantly changed our approach, and included many additional analyses. A brief description of the new methodology and results is provided below, followed by our detailed answers to all the questions raised by the reviewers. Reviewers' original questions/comments are in blue, and our answers are in black. In the revised manuscript, the revision is marked in red. Owing to the extensive changes made to the original manuscript, some comments on specific wording and styles may no longer be applicable. We will response to them accordingly.

Synopsis of the new methodology and results

This study aims to examine whether icebergs have a significant impact on the ocean net primary productivity (NPP) at the scale of the entire Southern Ocean (SO). We examine both large and small icebergs based on two separate datasets and determine if similar impacts can be observed. We first divided the SO into four ecological zones based on their different nutrient source and profile: the continental shelf zone (CSZ), the seasonal ice zone (SIZ), the permanent open ocean zone (POOZ), and the polar front zone (PFZ). Within each zone we compared NPP for grids with and without iceberg presence. For small icebergs, we found that grids with iceberg presence in general have higher NPP than those without icebergs. However, the impact is not uniform. In the CSZ where high level of iron is supplied through glacial meltwater and sediment input from the continent and continental shelf, and phytoplankton growth is largely limited by macronutrients, the presence of icebergs does not seem to have any impact on the ocean NPP. On the other hand, iceberg presence could significantly increase NPP in the high-nutrient low-chlorophyll (HNLC) regions. The NPP of grids with icebergs is 21% higher than those without in the SIZ, and 16% higher in the POOZ. The difference is slightly less (12%) for the PFZ where upwelling and eddy mixing could provide additional iron to the surface water, hence iron limitation is not as severe. Direct correlation between small iceberg frequency and NPP is weak although statistically significant. The strongest correlation is found at the SIZ which contains over 70% of the small icebergs by volume. For large icebergs, we found that the mean NPP for iceberg grids and their immediately adjacent grids is on average 10% higher than NPP of the nearby grids further away. However, the zonal response is different. The enhancement of NPP is the greatest in the high latitude zone of the CSZ and SIZ, which contains the majority of the large iceberg occurrences. Finally, we examined the secular trend of iceberg occurrence in the SO under the current climate change. For the entire period of the iceberg data, 1992-2014, both small and large icebergs have shown significant increasing trends. The increase is most rapid for the early 2000s, and has levelled off since then. For small icebergs, the increasing trend is most significant for the Pacific and Indian sections of the SO, whereas the Atlantic section of the SO shows no statistically significant trend. This could be related to the greater mass loss of the West Antarctica Ice Shelf, and the relative stability of the East Antarctica Ice Shelf under present climate change. For large icebergs, the increase is most significant for the Atlantic section, which contains 57% of all large icebergs for the study period. The reason for such disparity is unclear. As the climate continues to warm, the Antarctic Ice Sheet is expected to experience increased mass loss as a whole, which could lead to more

icebergs in the region. Based on our study, this could result in higher level of NPP in the SO as a whole, providing a negative feedback for global warming.

**Reviewer No. 1**

1) There seems to be a rather large disconnect in scale here that I'm having trouble understanding. It appears that most of the previous work on this subject has looked at iceberg properties and associated ocean/productivity effects at small scales – i.e. individual icebergs or icebergs within small defined areas. How do the authors justify scaling up to a hemispheric-scale analysis based on these limited findings?

Despite the great interests and several small scale areal studies suggesting increased concentrations of chlorophyll, krill, and seabirds surrounding icebergs, there seems to be a lack of large-scale studies that examine the impact of icebergs on total productivity in the SO. However, this is an important question, because in order for icebergs to have a significant role in the carbon cycle, hence serving as a possible negative feedback in the climate system, they should have an observable impact on NPP beyond the scale of individual icebergs. Therefore, this study aims to establish whether icebergs has a significant impact on marine productivity at the scale of the entire SO based on remote sensing data. In addition, most previous studies focus on large icebergs (> 18.5 km long) (Duprat et al. 2016, Helly et al. 2011). Although these large icebergs may have a big impact on productivity of surrounding area, they are relatively rare compared to much more abundant small icebergs. Tournadre et al. (2008) estimated over 8000 small icebergs (< 1km) alone in one year. Moreover, small icebergs have a much larger surface to volume ratio, providing relatively more substrate for organisms (Smith 2011). So when considered in total, they can have a big impact over much larger area in the SO. The majority of small icebergs come from the dislocation and breaking of large icebergs, and therefore serve as an important diffuse process for transport of freshwater and nutrients including iron (Tounadre et al. 2016). In this study, we examine both large and small icebergs to determine if similar impacts on the ocean net primary productivity (NPP) could be observed at the SO scale. In addition, this study also examines the secular trend of icebergs to establish whether increased icebergs could be observed with present climate change. These are important questions based on which we can evaluate whether icebergs could provide an important negative feedback for global warming.

References:

Duprat, L. P. A., Bigg, G. R., David J., and Wilton, D. J.: Enhanced Southern Ocean marine productivity due to fertilization by giant icebergs, Nature Geo., doi:10.1038/NGEO2633, 2016.

Helly, J. J., Kaufmann R. S., Vernet M., and Stephenson G. R.: Spatial characterization of the meltwater field from icebergs in the Weddell Sea. Proceedings of the National Academy of Sciences 108 (14) 5492-5497, doi:10.1073/pnas.0909306108, 2011.

Smith, K. L.: Free-drifting icebergs in the Southern Ocean: an overview. Deep Sea Research Part II: Topical Studies in Oceanography 58 (11): 1277-1284, doi:10.1016/j.dsr2.2010.11.003, 2011.

Tournadre, J., Whitmer K., and Girard-Ardhuin F.: Iceberg detection in open water by altimeter waveform
analysis. , J. Geophys. Res., Oceans 113, no. C8, doi: 10.1029/2007JC004587, 2008.

Tournadre, J., Bouhier, N., Girard-Ardhuin, F., and Rémy, F.: Antarctic icebergs distributions 1992–2014, J.
Geophys. Res., 121(1), 327-349, doi: 10.1002/2015JC011178, 2016.

One could imagine that icebergs from different regions would have radically different properties – volume, area, number, sediment concentration, sediment composition, etc. In particular, the bioavailable iron concentration would almost surely be different across various regions – how does one then compensate for that when scaling to the entire Southern Ocean?

Multiple sources of iron exist in the SO.  In the revised paper, in order to separate the impact of icebergs from some known sources, we divide our study area into four distinct ecological zones, each with different sources of nutrient supply (Fig. 1). These zones were adapted and modified from previous studies (Treguer and Jacques 1992, Moore and Abbott 2000, Ito et al. 2005), and within each zone the nutrient profile should be relatively similar. They are listed as follows:

- Continental shelf zone (CSZ): area less than 500 m in depth around the Antarctica continent and permanent ice shelves.
- Seasonal Ice Zone (SIZ): area north of the CSZ and within the mean maximum ice extent, defined by area with mean sea ice concentration greater than 70% in February. This region is delineated using the mean monthly sea ice 1979-2013 obtained from the National Snow and Ice Data Centre (ftp://sidads.colorado.edu/pub/DATASETS/nsidc0192_seaice_ trends_climo/monthly-climatology/).
- Permanent Open Ocean Zone (POOZ): area north of the SIZ and south of the Polar front zone (defined below). This zone is largely ice free throughout the year.
- Polar Front Zone (PFZ): area within 1 degree (~110km) of the polar front defined by Dong et al (2006). This is an area characterized by strong upwelling and eddy mixing.

The Subtropical Front, northern boundary of the Antarctica Circumpolar Circulation, usually defines the border of the SO. However, since few icebergs drift past the PFZ, areas of the SO north of the PFZ is not considered in this study.

References:

Tréguer, P. and Jacques, G.: Dynamics of nutrients and phytoplankton, and fluxes of carbon, nitrogen and silicon
in the Antarctic Ocean, Polar Biol., 149-162, 1992.

Moore, J. K. and Abbott, M. R.: Phytoplankton chlorophyll distributions and primary production in the Southern
Ocean, J. Geophys. Res., 105(C12), 28709-28722, doi: 10.1029/1999JC000043, 2000.

Ito, T., Parekh, P., Dutkiewicz, S., and Follows, M. J.: The Antarctic circumpolar productivity belt, Geophys.
Res. Lett., 32 (13), L13604, doi: 10.1029/2005GL023021, 2005.

[Figure]

Fig. 1: Ecological zones of the Southern Ocean.

**Impact of small icebergs on NPP**

Unlike large icebergs which are routinely tracked and monitored through satellite scatterometer instruments, small icebergs could not easily be identified until recently. The monthly dataset of small iceberg (<3km) is obtained from Iceberg Database of the Merged Altimeter for Altiberg project for 1992-2014 (ftp://ftp.ifremer.fr/ifremer/cersat/projects/altiberg/). This is the first and only dataset on small icebergs. The data is generated based on the analysis of high resolution altimeter waveforms from images of 9 satellite-based altimeters (Tournadre et al., 2016). The data contains three variables: iceberg presence probability, surface area and volume. Iceberg probability of presence is defined in Tournadre et al. (2012) as "the ratio of the number of icebergs detected within a grid cell by the total number of valid satellite data samples within the same grid cell". It is essentially normalized iceberg frequency. In the revised paper, we refer to this variable as such to make it more explanatory. Details of how iceberg surface area and volume are estimated can be found in Tournadre et al. (2012).

Figure 2a shows the spatial distribution of mean NPP (Fig. 2a) in the SO in relation to the defined ecological zones. Figure 2b shows the spatial distribution of normalized frequency for small icebergs in the SO in relation to the defined ecological zones.

[Figure]

Fig. 2. Spatial distribution of annual mean NPP (a) and normalized iceberg frequency (b) in relation to ecological zones.

For small icebergs, within each zone, we compared the mean NPP for grids with icebergs and those without. 
[revised manuscript text omitted]

2) The most recent work on the subject (Duprat et al., 2016) focused on large icebergs and their potential ocean impact. The authors make the assumption (lines 30-35) that the majority of small icebergs are associated with large ones, and therefore that total iceberg concentration should reveal similar impacts on ocean productivity. It's not clear to me what the basis is for this assumption. What is all or most of the sediment and hence available iron is limited to the large bergs? Smaller ones may simply be fragments that were never in contact with the bed surface. I'll readily admit here that I'm not an expert on iceberg processes, but this assumption is fundamental to the authors approach and it just seems unfounded to me.

Most previous studies focus on large icebergs (> 18.5 km long) (Duprat et al. 2016, Helly et al. 2011). Although these large icebergs may have a big impact on productivity of surrounding area, they are relatively rare compared to much more abundant small icebergs. Tournadre et al. (2008) estimated over 8000 small icebergs (< 1km) alone in one year. Moreover, small icebergs have a much larger surface to volume ratio, providing relatively more substrate for organisms (Smith 2011). So when considered in total, they can have a big impact over much larger area in the SO. The majority of small icebergs come from the dislocation and breaking of large icebergs, and therefore serve as an important diffuse process for transport of freshwater and nutrients including iron (Tounadre et al. 2016). Therefore, in this study, we examine both large and small icebergs to determine if similar impacts on the ocean net primary productivity (NPP) could be observed at the SO scale. Our results show that small icebergs indeed have a quite significant impact on the SO NPP.

Thank you for your helpful suggestions. We have redesigned our study and included many more new analyses, as explained in details above. Our results seem to suggest that both large and small icebergs are associated with elevated levels of NPP in the SO. However, the level of enhancement differs in different ecological zones, as we outlined above.

**Reviewer No. 2**

Wu & Hou use net primary production (NPP; estimated from satellite observations), temperature, and iceberg occurrence frequency data to investigate the variation of NPP with temperature and/or iceberg frequency in the Southern Ocean. They apply multi-linear regression (MLR) where the logarithm of NPP is considered as response to temperature and iceberg frequency; the logarithm of NPP is used to obtain residuals that better fulfil requirements for linear regression. The MLR and calculation of various coefficients of determination ($R^2$) is described in more detail than usual in the current literature showing the carefulness in application of these powerful methods. The authors claim that they found a small, however, statistically significant influence of icebergs.

The correlation coefficients ($r$) between the relative frequency of icebergs and (logarithm of) NPP and the corresponding coefficients of determination ($R^2$) based on MLR are small and would speak against an influence of icebergs on NPP. However, this negative result may be, at least in part, due to an inappropriate approach.

We agree that the MLR used in the original paper is not an appropriate approach. We significantly revised our study. First of all, multiple sources of iron exist in the SO. In the revised paper, in order to separate the impact of icebergs from some known sources, we divide our study area into four distinct ecological zones, each with different sources of nutrient supply (Fig. 1). These zones were adapted and modified from previous studies (Treguer and Jacques 1992, Moore and Abbott 2000, Ito et al. 2005), and within each zone the nutrient profile should be relatively similar. They are listed as follows:

- Continental shelf zone (CSZ): area less than 500 m in depth around the Antarctica continent and permanent ice shelves.

- Seasonal Ice Zone (SIZ): area north of the CSZ and within the mean maximum ice extent, defined by area with mean sea ice concentration greater than 70% in February. This region is delineated using the mean monthly sea ice 1979-2013 obtained from the National Snow and Ice Data Centre (ftp://sidads.colorado.edu/pub/DATASETS/nsidc0192_seaice_ trends_climo/monthly-climatology/).

- Permanent Open Ocean Zone (POOZ): area north of the SIZ and south of the Polar front zone (defined below). This zone is largely ice free throughout the year.

- Polar Front Zone (PFZ): area within 1 degree (~110km) of the polar front defined by Dong et al (2006). This is an area characterized by strong upwelling and eddy mixing.

The Subtropical Front, northern boundary of the Antarctica Circumpolar Circulation, usually defines the border of the SO. However, since few icebergs drift past the PFZ, areas of the SO north of the PFZ is not considered in this study.

[Figure]

Fig. 1: Ecological zones of the Southern Ocean.

**Impact of small icebergs on NPP**

Most previous studies focus on large icebergs (> 18.5 km long) (Duprat et al. 2016, Helly et al. 2011). Although these large icebergs may have a big impact on productivity of surrounding area, they are relatively rare compared to much more abundant small icebergs. Tournadre et al. (2008) estimated over 8000 small icebergs (< 1km) alone in one year. Moreover, small icebergs have a much larger surface to volume ratio, providing relatively more substrate for organisms (Smith 2011). So when considered in total, they can have a big impact over much larger area in the SO. The majority of small icebergs come from the dislocation and breaking of large icebergs, and therefore serve as an important diffuse process for transport of freshwater and nutrients including iron (Tounadre et al. 2016). Therefore, in this study, we examine both large and small icebergs to determine if similar impacts on the ocean net primary productivity (NPP) could be observed at the SO scale. Our results show that small icebergs indeed have a quite significant impact on the SO NPP.

Unlike large icebergs which are routinely tracked and monitored through satellite scatterometer instruments, small icebergs could not easily be identified until recently. The monthly dataset of small iceberg (<3km) is

obtained from Iceberg Database of the Merged Altimeter for Altiberg project for 1992-2014 (ftp://ftp.ifremer.fr/ifremer/cersat/projects/altiberg/). This is the first and only dataset on small icebergs. The data is generated based on the analysis of high resolution altimeter waveforms from images of 9 satellite-based altimeters (Tournadre et al., 2016). The data contains three variables: iceberg presence probability, surface area and volume. Iceberg probability of presence is defined in Tournadre et al. (2012) as "the ratio of the number of icebergs detected within a grid cell by the total number of valid satellite data samples within the same grid cell". It is essentially normalized iceberg frequency. In the revised paper, we refer to this variable as such to make it more explanatory. Details of how iceberg surface area and volume are estimated can be found in Tournadre et al. (2012).

Figure 2a shows the spatial distribution of mean NPP (Fig. 2a) in the SO in relation to the defined ecological zones. Figure 2b shows the spatial distribution of normalized frequency for small icebergs in the SO in relation to the defined ecological zones.

[Figure]

Fig. 2. Spatial distribution of annual mean NPP (a) and normalized iceberg frequency (b) in relation to ecological zones.

For small icebergs, within each zone, we compared the mean NPP for grids with icebergs and those without. 
[revised manuscript text omitted]

Wadley, Martin R., Timothy D. Jickells, and Karen J. Heywood. "The role of iron sources and transport for Southern Ocean productivity." Deep Sea Research Part I: Oceanographic Research Papers 87 (2014): 82-94.

**Impact of large icebergs on NPP**

Large icebergs are routinely tracked and monitored. The Brigham Young University (BYU) Centre for Remote Sensing produces and maintains an Antarctica Iceberg Tracking Database (http://www.scp.byu.edu/data/iceberg/database1.html) for icebergs with length larger than 6 km (Stuart and Long 2011) since 1992, using six different satellite scatterometer instruments. Icebergs are identified using enhanced resolution scatterometer backscatter images. The dataset contains the daily location for all identified icebergs. We summarize the track data into monthly 1x1 degree gridded format to facility the analysis with NPP data. Large icebergs are relatively rare. For the period 2002-2014, 393 icebergs were identified in the BYU data set. Among them, 154 are icebergs larger than 18.5 km in length. They are named icebergs monitored by the National Ice Centre. The other 239 are smaller icebergs between 6 and 18.5 km. Their tracks are presented in Fig. 3a, and the summarized gridded count data in Fig. 3b. The majority of large icebergs concentrate near the coastal region of Antarctica, where they are calved from the major ice shelves, such as the Ross, Filchner, Ronne, Larsen, and Amery. Large number of icebergs are also present in the south Atlantic section of the SO, originated mostly from the ice shelves in the Weddell Sea.

[Figure]

Fig. 3: Spatial distribution of large icebergs. (a) all iceberg tracks between 2002 and 2014. (b) the gridded count of iceberg occurrences between 2002 and 2014.

Since large icebergs are relatively rare, at certain point in time, they cover very small portion of the ocean surface. It is therefore not appropriate to adopt the same approach used in studying small icebergs. We use a different approach instead. We first summarize the track data into monthly snapshots of 1x1 degree grid of iceberg counts. The great majority of the grids only gets 1 large iceberg. Very rarely were more than one large iceberg present in a grid at a given month. For each iceberg occupied grid, we select a 7x7 degree (49 grids) window around it. Within the window, we identify all grids with icebergs (iceberg grids), grids immediately adjacent to iceberg grids (adjacent), and the rest of the grid within the window (nearby grids). We calculate the mean NPP for each group for a single iceberg grid, and repeat the same operation for all iceberg grids at monthly timescale. We then compare the mean NPP of these three groups to see if they are significantly different. Pair-wise t test is used to establish the statistical significance of the difference between groups. The difference is summarized by ecological zones, and their seasonal variations examined.

Based on this methodology, we calculated the mean NPP each of the three groups at monthly timescale. Results are summarized by ecological zones (Table 2). Most of the iceberg grids are located in the CSZ and the SIZ, but their frequency is still relatively low, at 73 and 60 grids per year respectively. In general, the difference in mean NPP between iceberg grids and adjacent grids are fairly small, but the mean NPP of combined iceberg and adjacent grids are significantly (about 10%) higher than the nearby grids. This pattern is fairly consistent in most of the ecological zones with the only exception of the POOZ, where iceberg/adjacent grids have similar NPP as the nearby grids. Seasonally, the most significant increase of NPP from surrounding grids occurs in the most productive months in the austral summer (December, January and February). The general enhancement of NPP near icebergs is consistently found with both small and large icebergs, although the zonal response is different. With small icebergs, the largest enhancement is seen in zones relatively poor in iron such as the POOZ and the

SIZ, whereas the effect in the CSZ is not significant. The large icebergs, on the other hand, seem to increase NPP more in the higher latitudes such as the CSZ and the SIZ. However, the low frequency of large icebergs makes it difficult to establish reliable statistical relationship. Moreover, the coarse spatial resolution (getting even coarser at lower latitudes of the POOZ and PFZ) could make it harder to detect enhancement of NPP near large icebergs.

| Zone | Mean NPP of iceberg grids (mgC m$^{-2}$ d$^{-1}$) | Mean NPP of adjacent grids (mgC m$^{-2}$ d$^{-1}$) | Mean Diff. | p value | Mean NPP of iceberg and adjacent Grids (mgC m$^{-2}$ d$^{-1}$) | Mean NPP of nearby grids (mgC m$^{-2}$ d$^{-1}$) | Mean diff. | p value | No. of Iceberg grids per year |
|------|------|------|------|------|------|------|------|------|------|
| CSZ | 241.41 | 240.79 | 0.62 | 0.84 | 241.54 | 214.62 | 27.17 | 0.00 | 73.08 |
| SIZ | 206.05 | 210.66 | -4.61 | 0.05 | 208.74 | 188.72 | 20.03 | 0.00 | 60.92 |
| POOZ | 151.13 | 148.93 | 2.19 | 0.25 | 149.27 | 149.14 | 0.13 | 0.94 | 14.23 |
| PFZ | 248.78 | 257.69 | -8.91 | 0.02 | 255.72 | 242.66 | 13.06 | 0.02 | 3.77 |
| Total | 219.15 | 220.87 | -1.72 | 0.33 | 220.39 | 199.17 | 21.31 | 0.00 | 152 |

Table 2: Mean NPP of iceberg grids, adjacent grids and nearby grids.

By considering the whole region south of 40 S, the authors include large areas that never see an iceberg and where thus icebergs cannot really influence NPP.

In the revised paper, we limited our study to the area of the SO south of the polar front zone, as very few icebergs drift past this zone. See above for more details.

On the other hand, temperature correlates with almost everything and is, at least on the large scales considered here, also a proxy for latitude, light and maybe other quantities. I'm convinced that large icebergs can have an influence on NPP, however, the question is which mechanisms are at work here (iron supply, upwelling of freshwater, increased mixing) and, depending on the mechanism, how large is the area of influence.

We no longer use regression model in the revised study, and temperature is hence not examined as a factor for NPP and iceberg variation. Our revised study seems to suggest that icebergs are associated with elevated levels of NPP in the SO. See above for more details.

I suggest that the authors look at their results with open mind and discuss limitations of their approach in the light of known as well as speculative mechanisms.

Thank you for your suggestions. We have redesigned our study with new methodology and many new analyses, based on which possible mechanism and limitations are discussed. See above for details.

General comments:

The terms 'iceberg probability of presence' and 'iceberg presence probability' should be avoided. I would prefer 'relative frequency of icebergs'.

Iceberg probability of presence is defined in Tournadre et al. (2012) as "the ratio of the number of icebergs detected within a grid cell by the total number of valid satellite data samples within the same grid cell". It is essentially normalized iceberg frequency, and we shall refer to this variable as such to make it more explanatory.

Reference:

Tournadre, J., Girard-Ardhuin, F., and Legrésy, B.: Antarctic icebergs distributions, 2002–2010, J. Geophys. Res., 117, C05004, doi:10.1029/2011JC007441, 2012.

The interpretation of correlation coefficients depends very much on the context (for example, high-quality measurements in branches of physics versus ecological observations with small sample sizes). A rule of thumb might be 'no or weak correlation' for $-0.3 < r < +0.3$, 'positive correlation' for $r > 0.3$. What's your interpretation of correlation coefficients? What is meant by 'significant' in this context?

Regarding your question, in our original paper, "significance" is established based on statistical tests. A correlation is deemed significant if the probability of it being a result of random variation is low (i.e. small p value). A correlation can be statistically "significant" even when the coefficient is low, if it is derived from large number of data points with relatively low variance.

That being said, we agree that the direct correlation between NPP and iceberg frequency is low. Therefore, we designed new ways other than correlation to examine the impact of icebergs on NPP (see above). However, in the revised paper, we did include a section on the correlation between NPP and small iceberg frequency, and discussed possible reasons for low correlation. The frequency of large icebergs is too low for a robust correlation analysis.

**Correlation between NPP and small iceberg frequency.**
In order to further examine the quantitative relationship between NPP and icebergs, we conduct the correlation analysis between the two factors. Normalized iceberg frequency is chosen over iceberg volume because it seems to better correlate with NPP. The possible reason is that for large icebergs basal melting is small compared to their breaking into smaller icebergs (Tournadre et al. 2015). These smaller icebergs act as an important diffuse process for nutrient transport. Therefore, iceberg frequency may have a more direct impact on NPP than total iceberg volume. We calculate Pearson's correlation coefficient (r) for annual total net production and the annual mean iceberg frequency. The use of annual data eliminates the seasonal cycles that exist in both variables, which can artificially inflate the significance of correlation. Given that both variables are positively skewed, we also tried non-parametric rank correlation Spearman's rho. Both methods yielded similar results, with Spearman's rho giving slightly higher correlation than Pearson's r, indicating the existence of non-linear correlation. We perform the correlation analysis in two ways. First, we calculate the correlation coefficient between annual production and iceberg frequency at each grid point (local correlation), so that we can control it for spatially varied factors such as solar radiation, length of day, and ocean circulations. We then summarize the correlation coefficients by

ecological zones. Second, we calculate a single correlation coefficient for annual productivity and iceberg frequency of all grid points within a zone (zonal correlation), and compare the strength of correlation between zones.

Results of local correlation between annual NPP and iceberg frequency at each grid points are presented in Fig. 4 and summarized in Table 3. Only Pearson's r is reported, as Spearman's rho gives similar results.

[Figure]

Fig. 4. Temporal correlation between annual NPP and normalized iceberg frequency.

[revised manuscript text omitted]

When considering the whole oceanic area south of 40 S, many 1 x 1 cells have very low relative frequencies of icebergs and any variation of NPP in these 'low frequency' cell cannot be 'explained' by icebergs. Thus it is not surprising that correlation with iceberg frequency is low (r = -0.03). If restricting the area to cells where the relative frequency of icebergs is larger than zero (is zero really the threshold value), it is not surprising that the value of the correlation coefficient changes, however, r = 0.12 is still very small (-> rˆ2 = 0.01!!!). I don't understand what is meant by 'when temperature is controlled'. A correlation coefficient of 0.27 is in my opinion a borderline case.

We agree that the approach adopted in the original manuscript is not appropriate. In the revised paper, we completely redesigned the study (see above).

p.6 'However, the effect of the iceberg probability on NPP increases as measured by both R2 (0.02) and standardized coefficient (0.15). This effect is statistically significant at critical level of 0.01.' I don't understand what the authors would like to convey here. R2 = 0.02 is small and thus iceberg frequency is not a good quantity for predicting or explaining variations in NPP. I do not know how the authors calculated a p-value below 0.01 and what it means in the current context. The conclusion is mainly based on these numbers (R2 = 0.02, p < 0.01):

"… our analyses show that iceberg presence has a small, yet statistically significant, positive impact on the SO NPP. … in places with iceberg presence, iceberg probability could independently explain 2% of the NPP'.

We agree that the approach adopted in the original manuscript is not appropriate. In the revised paper, we completely redesigned the study (see above).

Fig.2: I doubt that comparison of zonally averaged NPP & relative frequency of icebergs yields much insight (please drop figure).

This figure is deleted.

The text needs a bit polishing by a native English speaker (examples: 'planktons', 'which is much contrasted in the three ocean basins')

The above mentioned grammatical mistakes have been corrected. The revised text has been carefully checked for grammatical errors.

Specific comments:

- Southern Ocean is defined in the manuscript as the oceanic region south of 40S (which is fine with me): you don't have to repeat this definition several times

We have paid attention to exclude such repetitions in our revised paper.

- abstract: 'NPP in the SO is largely influenced by temperature' I suggest reformulation because MLR only shows variation of NPP with temperature and not (direct) 'influence'. Temperature is co-varying with many other quantities in the SO and thus it is not clear by what mechanism NPP is 'influenced' by temperature.

Owing to the extensive revision of the paper, the abstract is now completely different. This sentence is no longer in the abstract.

- page1, line 27: planktons -> plankton

Corrected.

- page1, line 30: "in either natural or artificial settings" you might cite here: Blain, S., Quéguiner, B., Armand, L., Belviso, S., Bombled, B., Bopp, L., ... & Christaki, U. (2007). Effect of natural iron fertilization on carbon sequestration in the Southern Ocean. Nature, 446(7139), 1070-1074.

- Smetacek, V., C. Klaas, V.H. Strass, P. Assmy, M. Montresor, B. Cisewski, N. Savoye, A. Webb, J.M. Arrieta, U. Bathmann, R. Bellerby, G.M. Berg, P. Croot, F. d'Ovidio, S. Gonzalez, J. Henjes, G.J. Herndl, L.J. Hoffmann, H. Leach, M. Losch, M.M. Mills, C. Neill, I. Peeken, R. Röttgers, O. Sachs, E. Sauter, M.M. Schmidt, J. Schwarz, A. Terbrüggen, & D. Wolf-Gladrow, Deep carbon export from a Southern Ocean iron-fertilized plankton bloom, Nature, 487, 313-319, 2012. doi:10.1038/nature11229

These new citations are included in the revised manuscript.

p.2 total dissolved Fe in SO: you cite more recent work: Klunder, M. B., Laan, P., Middag, R., De Baar, H. J. W., & Van Ooijen, J. C. (2011). Dissolved iron in the Southern Ocean (Atlantic sector). Deep Sea Research Part II: Topical Studies in Oceanography, 58(25), 2678-2694.

Klunder, M. B., Laan, P., De Baar, H. J. W., Middag, R., Neven, I., & Van Ooijen, J. (2014). Dissolved Fe across the Weddell Sea and Drake Passage: impact of DFe on nutrient uptake. Biogeosciences, 11(3), 651-669.

These new citations are included in the revised manuscript.

p.2, lines 9-11: Fe from sediments has to be mixed up or upwelled; Fe source from hydrothermal vents is missing.

German, C. R., Legendre, L. L., Sander, S. G., Niquil, N., Luther, G. W., Bharati, L., ... & Le Bris, N. (2015). Hydrothermal Fe cycling and deep ocean organic carbon scavenging: Model-based evidence for significant POC supply to seafloor sediments. Earth and Planetary Science Letters, 419, 143-153.

This has been added to the revised manuscript.

p.2, line 16: drop 'Thus'

Deleted.

p.2 'Raiswell and Canfield (2012) recently even suggested that icebergs could supply more than 90% of total colloidal and filterable Fe in the SO.' Raiswell and Canfield (2012) write: 'The model indicates that the rate of delivery of bioavailable Fe from icebergs to the Southern Ocean is at least as large as that by wind-blown dust. However estimates of all the main aqueous, nanoparticulate and colloidal (and potentially bioavailable) Fe inputs to the ocean are poorly-constrained.'

We have incorporated this in our Introduction section in the revised paper.

p.4, lines 8-9 'is the mean of the variable and is the standard deviation of the variable' -> 'm is the mean and s is the standard deviation of the sample' & change eq. accordingly, i.e. $z = (x - m)/s$

This section has been deleted from the revised paper.

p. 4, lines 26-27 'NPP is relatively high near the coast of Antarctica, largely because of nutrient input from the continent.' I suggest replacing 'nutrient' by 'iron'.

It has been changed accordingly.

p.4 -60S -> 60S [drop minus sign: S already indicates 'negative' latitudes; no space between degree symbol and S]; please change everywhere in manuscript

This has been corrected throughout the revised manuscript.

---

## Author Response (AR2)

Dear Editor,

Thank you for your helpful comments. Below are our responses. The original comments are in black, and our responses are in blue. The changes made in the manuscript are also marked in blue.

We look forward to hearing back from you.

Best regards,
Shuang-Ye Wu

Comments to the Author:
This revised manuscript is much clearer and the revisions have addressed many of the reviewer comments. After reading the paper I do think there are some additional issues that remain to be addressed, listed below. The most substantive comment is the last one, concerning separation of the observation of weak association of icebergs with NPP from hypotheses about functional relationships.

1) On page 6, lines 10-15, the F test for linear trend in iceberg changes is mentioned. Although the F test is fairly standard a reference here would be helpful.
The following reference is added:
Lomax, Richard G. (2007). Statistical Concepts: A Second Course. p. 10. ISBN 0-8058-5850-4.

2) In the tables, where means are presented, and p values given, a measure of uncertainty in the means (for example 95% confidence interval) should also be given - really, for all reported statistics in this paper the confidence interval should be given. Where p values are given the test these are relevant to should be mentioned in the figure caption as well.
Standard errors are calculated for the mean values and added to the tables and in the paper to indicate the uncertainty. Statistical tests are added as notes for all p-values reported in the tables.

3) On page 6, lines 29-31, NPP differences between cells with and without icebergs are discussed for seasonal data. Although there are differences they are very small, are these really important?
Seasonal NPP differences between cells with and without icebergs are discussed on page 7, lines 25-31. Although the differences are small, they have seasonal variations. The differences are usually larger in the productive months (October – March), averaging 7-10%, reaching 15-20% in some months in some zones. We believe that this difference is not insignificant, and the seasonal and zonal variation are important patterns. Therefore, we decided to keep the original discussion.

4) Page 10, lines 20-26. This section starts by saying that the general enhancement of NPP near icebergs is consistent but then says that the small iceberg effect in the CSZ is not significant. These seem contradictory statements to me. The large iceberg effect in the PFZ also seems to be ignored in this section, and then the section ends by saying the data may be too coarse. So, I am the first sentence of this section seems at odds with its remainder - and my suggestion is to reword the first part unless you really feel you can demonstrate a consistent enhancement.

We reworded the first sentence as follows:
*"Enhanced levels of NPP are often found with both small and large icebergs, although their zonal response is different."*

5) Page 10, lines 28-32. Significance is repeated several times here, this could be reworded.
These lines has been reworded as follows:
*"Over the period of 1992-2014, the amount of small icebergs has a rather notable increasing trend (Figure 7). Annual iceberg volume increases at 2.6% per year, and iceberg frequency increases at 4.7% per year. Both trends are statistically significant (Table 5). Iceberg amount increases the most for the period 1992-2004, and decrease slightly since then. In particular, there seems to be a period of rapid increase in the first half of the 2000s. Large iceberg frequency, measured as the mean annual number of grids occupied by large icebergs, follows a similar increasing trend (Figure 8) at 7.09% per year (Table 6)."*

6) Page 11, line 5, type "ad"
"ad" is changed to "and".

7) Page 11, line 11-12. This sentence seems to be saying that because icebergs did not change much then it is not surprising that NPP did not change? But the alternative hypothesis is that icebergs don't matter to NPP, or that other things were affecting NPP, given that the relationships presented elsewhere in the paper are not that strong. I think this could be reworded, perhaps to say that if icebergs are important to NPP then it is not surprising that a trend in NPP was not found.
This has been reworded as suggested:
*"If icebergs play an important role in enhancing the SO NPP, it is probably not surprising to see the lack of any observable trend in the total NPP of the SO for the data period of 2002-2014 (Figure 9), given the slight negative to no trends in iceberg presence for the same time period."*

8) Page 12, line 26-29. It is first stated that in general grids with more icebergs have high NPP but then an exception (the CSZ) is noted. "In general" implies to me a consistent pattern, rather than one with clear exceptions, so I suggest rewording.
The sentence has been reworded:
*"We found that in many places grids with iceberg presence have higher NPP than those without icebergs."*

9) In the conclusions section I am concerned about concluding there is a functional relationship between icebergs and NPP based on the correlations in the data. Reviewers also brought this up. For example, the statements that "Therefore, icebergs, large and small, have an observable positive impact on ocean NPP at the SO scale" and that that icebergs might provide a negative feedback to warming imply that the data presented here prove there is a functional relationship, but as I understand what they show is there is a statistical association, mostly fairly weak. Readers who do not look carefully at the data in the paper may draw the wrong conclusion from these statements so I suggest some revision is needed to carefully separate the observations from hypotheses about what they mean. There are similar statements in other places in the manuscript.

In the revised manuscript, we reworded our conclusion to tone down any suggestion of "functional relationship" between icebergs and NPP. For example, in many places, we replaced the word "impact" with "association". The revised conclusion is as follows:

*"This study aims to examine whether icebergs have a significant impact on the ocean NPP at the scale of the SO. Through using remote sensing data, we examine the impacts of both small and large icebergs on the ocean NPP. We divided the SO into four ecological zones based on their different nutrient source and profile. For small icebergs, we compared NPP for grids with and without iceberg presence within each zone. We found that in many places grids with iceberg presence have higher NPP than those without icebergs. However, the impact is not uniform. In the CSZ where high level of iron is supplied through glacial meltwater and sediment input from the continent and continental shelf and phytoplankton growth is largely limited by macronutrients, the presence of icebergs does not seem to have any impact on the ocean NPP. On the other hand, Iceberg presence is associated with significantly higher NPP values in the HNLC regions. The NPP of grids with icebergs is 21% higher than those without in the SIZ, and 16% higher in the POOZ. Direct correlation between iceberg frequency and NPP is weak although statistically significant. Strongest correlation is found at the SIZ, which contains over 70% of the icebergs by volume. For large icebergs, we examine the average NPP of iceberg occupied grid cells, immediately adjacent cells, and nearby cells that are further away. We found that NPP of iceberg cells and adjacent cells is on average 10% higher than NPP of nearby cells. The positive impact of large icebergs is stronger in high latitude zones of the CSZ and the SIZ, where most of them occur. Therefore, icebergs, large and small, have an observable positive association with ocean NPP at the SO scale. For the entire period of the iceberg data, 1992-2014, both large and small icebergs have shown significant increasing trends. The increase is most rapid during the first half of 2000s, and levels off since then. For small icebergs, the increasing trend is most notable for the Pacific and Indian sections of the SO, whereas the Atlantic section of the SO shows no statistically significant trend. This could be related to the greater mass loss of the West Antarctica Ice Shelf, and the relative stability of the East Antarctica Ice Shelf under present climate change. The sectional trends are different for large icebergs, which increase significantly for the S. Atlantic and S. Indian sections, but remain relatively unchanged in the S. Pacific section of the SO. The very low frequency of icebergs in the S. Pacific section of the SO makes it harder to detect any trend. However, the exact mechanism that accounts for the difference in sectional trends between small and large icebergs remains unclear. As the climate continues to warm, the Antarctic Ice Sheet is expected to experience increased mass loss as a whole, which could lead to more icebergs in the region. Based on the positive association between icebergs and NPP shown in this study, this could result in higher level of NPP in the SO as a whole, providing a possible negative feedback for global warming."*

---

## Author Response (AR3)

Dear Editor,

Thank you for your further comments. Below are our responses. The original comments are in black, and our responses are in blue. The changes made in the manuscript are marked in red.

We look forward to hearing back from you.

Best regards,
Shuang-Ye Wu

Thank you for considering all of my comments and for your clear responses. I have some small mostly technical comments on the revised manuscript but would ask that you consider point #2 below particularly.

1) Abstract, lines 16-19 – the wording still, to me, strongly implies causation ("icebergs increase NPP") but as discussed in other comments what is shown is an association in the data.

We "softened" our language to focus more on association, rather than causation. The revised language is as follows:

"We found that the presence of icebergs is associated with elevated levels of NPP, but the difference varies in different zones. Grid cells with small icebergs on average have higher NPP than other cells in most iron deficient zones: 21% higher for the SIZ, 16% for the POOZ, and 12% for the PFZ. The difference is relatively small in the CSZ where iron is supplied from melt water and sediment input from the continent. In addition, NPP of grid cells adjacent to large icebergs on average is 10% higher than that of control cells in the vicinity. The difference is larger at higher latitudes, where most large icebergs are concentrated."

2) Abstract, lines 22-23. About the negative feedback on global warming. Because this would be an important point for carbon cycle science, I would like to ask if you can quantify the potential impact based on the data in the paper. For example, what is the total difference in NPP integrated across the different zones for regions with and without icebergs? Is it at all significant relative to the total global NPP? If not, can the statement about negative feedback be justified? Warming may produce more icebergs, but eventually it would produce less, as ice margins retreat on to land. I have the same comment about the text in the conclusions.

Based on the data used in this paper, it is possible to provide a quantitative estimate on the potential impact of the iceberg on NPP, although with great uncertainties. At the moment, it is a relatively small quantity of additional carbon (~0.05 Gt) produced because of the presence of icebergs. How much this could increase with global warming remains uncertain. However, this may not be insignificant in terms of global carbon cycle. Although global ocean NPP exceeds 100 Gt, most of it goes back to the atmosphere after organisms die. The Southern Ocean, on the other hand, is a place with major downwelling. As a result, the additional carbon could be transported to and stored in deep ocean for much longer period of time. Therefore, the possible increase of icebergs with global warming at least in the near future could potentially provide a negative feedback. We added "in the near future" in the abstract and conclusion to indicate the time frame that this possible feedback could work.

3) Page 3, line 2. "Has" should be "have."

This was corrected.

4) Page 3, line 18. Here and elsewhere, data is a plural word, please check the text.

"Data" was changed to a plural word throughout the manuscript.

5) Page 7, line 25. Are all differences significant? CSZ differences are not, correct? Can you give differences and p values in Table 2?

The main text was changed to "With the exception of the CSZ, the differences in all other zones are statistically significant based on the two-sample Student's t test." The differences are added to Table 2 with their statistical significance indicated.